# Evaluation of Alternative-to-Gas Chlorination Disinfection Technologies in the Treatment of Maltese Potable Water

**Georgios Psakis** [1,*] , **David Spiteri** [2], **Jeanice Mallia** [2], **Martin Polidano** [2], **Imren Rahbay** [2]
**and Vasilis P. Valdramidis** [1,3]

1   Faculty of Health Sciences, Department of Food Sciences and Nutrition, University of Malta, Mater Dei Hospital, MSD 2080 Msida, Malta; vasilis.valdramidis@um.edu.mt
2   Water Services Corporation (WSC), Triq Hal Qormi, LQA 9043 Hal Luqa, Malta; david.spiteri@wsc.com.mt (D.S.); jeanice.mallia@wsc.com.mt (J.M.); martin.polidano@wsc.com.mt (M.P.); imrenrahbay@gmail.com (I.R.)
3   Department of Chemistry, National and Kapodistrian University of Athens, 157 84 Zografou, Greece
*   Correspondence: georgios.psakis@um.edu.mt

**Abstract:** For years, gas chlorination has been the adopted disinfection technology in the treatment of Maltese potable water. Despite its strong bacterial inactivation potential, traditional chlorination generates high free chlorine residual and disinfection by-products that compromise the sensory attributes of drinking water and deter the population from consuming it. We have identified plausible alternative-to-gas-chlorination technologies for its treatment, with the aim of (a) reducing the disinfectant and/or chlorination dose used for microbial inactivation, and (b) attenuating the negative impact of putative disinfection by-products on the water's organolepsis, while safeguarding its safe-for-consumption characteristics. We have subjected ultraviolet C (UVC) irradiation, hydrodynamic cavitation (HC), $ClO_2$ generation, and electrochlorination (NaClO) to bacteriological and physicochemical bench-scale studies to assess their bacterial inactivation efficacy and by-product generation propensity, respectively. All the tested technologies except HC achieved a minimum of 3 $Log_{10}$ microbial inactivation, with NaClO and $ClO_2$ appearing more effective over neutral and alkaline pH conditions, respectively. In addition, we have identified synergistic effects of cavitation on UVC for *Enterococcus faecalis* inactivation, stemming from enhancement in oxidative stress. Moderate reductions in the total dissolved solid content and $Ca^{2+}$ hardness of the tested water also emerged following prolonged cavitation. For feasibility studies, the performance of the technologies was further evaluated on the following areas: (a) implementation, (b) practicality, (c) adaptability, (d) integration, (e) environment and sustainability, and (f) cost and effect. Electrochemical generation of NaClO emerged as the most promising technology for further on-site work, followed by $ClO_2$ and UVC.

**Keywords:** ultraviolet light; hydrodynamic cavitation; chlorine dioxide; electrochlorination; disinfection by-products; Maltese tap water

## 1. Introduction

Maltese potable water is a blend of sea water desalinated by reverse osmosis (RO), and groundwater obtained from the perched and mean sea level aquifers of the islands [1]. Historically, initial blends comprised ground to RO water at a 6:4 ratio. However, (a) quality deterioration of groundwater resources and (b) excessive water demand posed by the continuously increasing population density of the islands have necessitated blending practices with 60% RO water [2]. Prior to blending, RO water is subjected to lime treatment for raising its pH and reducing the magnesium hardness [3].

For over 100 years, traditional chlorination ($Cl_2$ gas, Text S1) has been the adopted disinfection method of RO-softened and borehole water bodies before reaching the reticulation

networks, to ensure the delivery of safe water for consumption. Despite its decontamination effectiveness, gas chlorination generates high free available chlorine and several disinfection by-products (DBPs; e.g., trihalomethanes (THMs), Section 2.7), which not only raise health concerns [4] but also compromise the organoleptic properties of the water and challenge the consumer's sensory experience [5].

Out of the necessity to minimize the impact of residual chlorine and DBPs on the organoleptic properties of Maltese drinking water and generate a product that adheres to the standards of the EU directive, whilst satisfying consumer sensory perceptions, we sought for chlorine-gas-alternative disinfection technologies. Of the available alternative disinfection processes (physical, chemical, or membrane-based) we selected:

1.  Ultraviolet-light C (UVC) irradiation; the antimicrobial properties of UVC (254 nm) have been demonstrated for a wide range of pathogens [6]. UVC can be easily implemented both at the level of water transfer from the galleries/boreholes to the reservoir, as well as before release to the distribution and reticulation networks, and despite its lack of residual disinfection, it can be combined with chemical processes to achieve greater microbial $Log_{10}$ reductions in a cost-effective and eco-friendly manner [7]. Additionally, UVC-based treatments contribute negligibly to DBP formation and were shown to attenuate the toxicity of chlorine-derived DBPs [8], thus leaving the water's organoleptic characteristics unaffected.

2.  Hydrodynamic cavitation (HC); HC has been emerging as an effective technology of water treatment [9]. HC not only can achieve bacterial reductions in the range of 0.6–5 $Log_{10}$ without generating DBPs but is also a cost-effective and sustainable method for water softening [10]. It can work synergistically with UVC in the attenuation of dissolved oxygen carbon under advanced oxidation processes [11] and can be combined with electrochlorination for the attenuation of chloroform [12].

3.  Chlorine dioxide generation; $ClO_2$ (generated in situ) provides an attractive alternative to chlorination as it: (a) is effective at low concentrations (<1 mg/L); (b) is active over a broad pH range (4–10); (c) has a long residual activity; and d) generates fewer harmful DBPs, thus impacting less on the water's organolepsis [13]. It can be combined with other disinfection technologies, like UVC, to achieve higher microbial reductions [14], and to attenuate DBP formation [15].

4.  Electrochlorination (NaClO); electrochemically generated sodium hypochlorite will dissociate to hypochlorous acid with subsequent degradation to chlorate and chloride. Though not fundamentally different to gas chlorination, electrochlorination is an attractive technology because: (a) desired end concentrations of active oxidants are generated on site without the need of chlorine gas, thereby reducing transportation and storage necessities and minimising leakage risks [16], (b) it generates less haloacetonitriles than chlorine gas [16], (c) it can remain longer in the distribution system for effective biofilm formation control [16] and (d) it can be combined with physical and chemical disinfection to attenuate DBP formation (Table S7) [15,17–30].

Here we present the outcomes of undertaken bench-scale studies to assess the decontamination efficiency of the selected technologies in the treatment of Maltese potable water using *Escherichia coli* and *Enterococcus faecalis* as quality indicators. We are also presenting a small-scale feasibility study with the aim of selecting the best three technologies for implementation in future plant-pilot work.

## 2. Materials and Methods

### 2.1. Equipment and Bench-Scale Study Configurations

2.1.1. UVC Set-Up

For bench-scale studies, the UV-405 LCD system (Sita: Arnheim, The Netherlands) was employed. The system comprised a 30 W UVC lamp (254 nm), providing a maximum dose of 300 J/m$^2$ for flow rates up to 20 L/min, with an LCD plus unit equipped with a detector for monitoring intensity fluctuations over 4–20 mA contact. Similar systems have been reported to achieve a minimum of 3 $Log_{10}$ microbial inactivation for laminar flows of

10–100 L/min [7]. To determine the disinfection efficiency of the system over different flow rates, in addition to gravity (flow rates of 0.1–1 L/min), two different pumps were used to extend the range of testable flows to 15 L/min (Figure 1a). Flow rates were monitored by a digital flow meter (DigiFlow 6710M-44, New York, NY, USA) (Figure 1a).

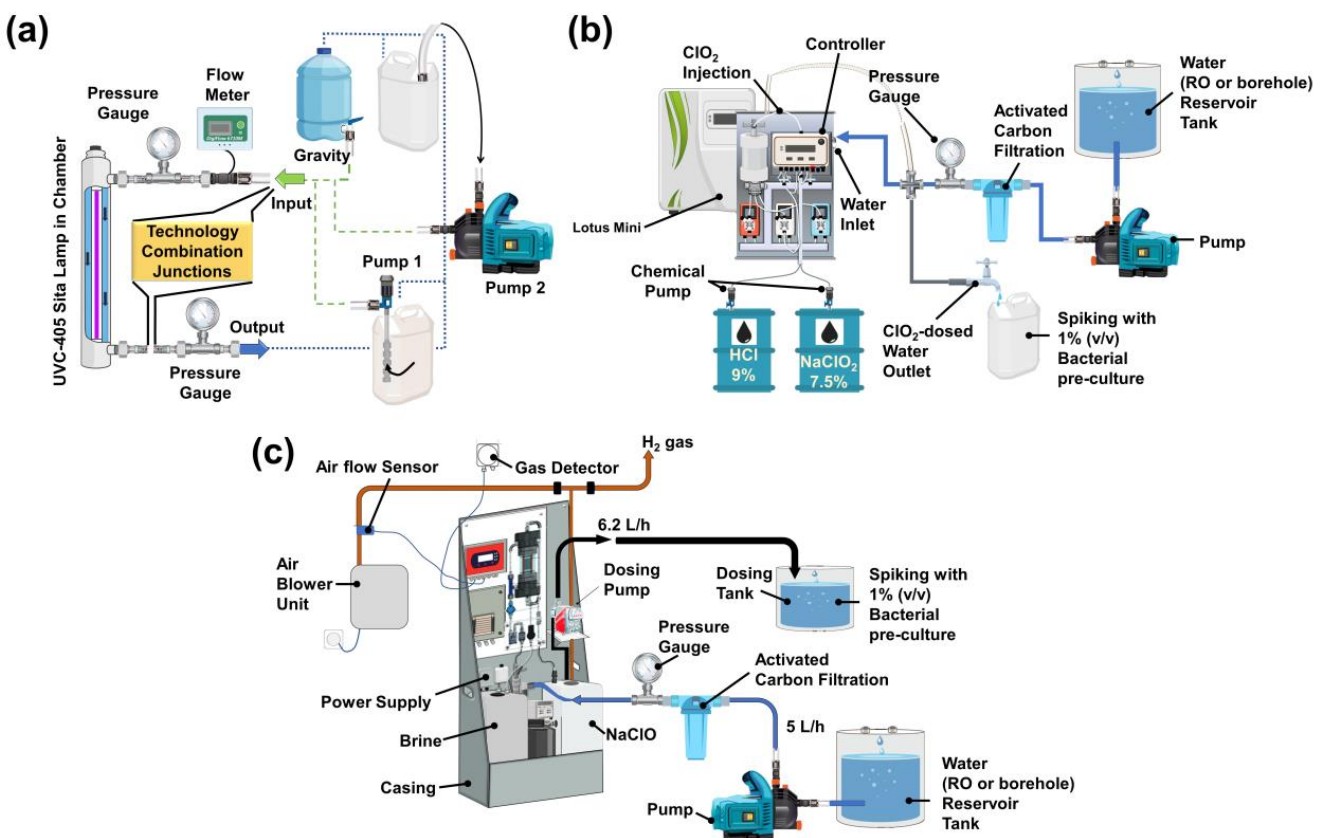

**Figure 1.** Bacterial disinfection configurations employed in bench-scale studies. (**a**) UVC set-up; input water is delivered to the UVC chamber (green dashed lines) through gravity or pumps (pumping denoted with solid black arrows). Input flow rates can be determined by a flow meter. UVC-treated water exits the chamber (output) before being collected in its original containers for subsequent UV-exposures (dotted blue lines). Samples for analysis were collected following the treatment of each litre of water. HC can replace or be integrated upstream or downstream from the UVC device for evaluation of performance in hybrid schemes. (**b**) $ClO_2$ set-up; carbon-filtered RO/borehole water is fed into the Lotus Mini $ClO_2$ generator (10–20 L/h at an inlet pressure controlled by a pressure gauge) and is used to drive the in situ generation of $ClO_2$ from HCl and $NaClO_2$. Generated $ClO_2$ is collected and is used to disinfect spiked water for the initiation of inactivation studies. (**c**) Electrochlorination set-up; in situ generation of NaClO is electrolytically driven from water-dissolved brine. Generated NaClO is collected in the designated tank and a dosing pump is used to treat the desired water source.

### 2.1.2. HC and UVC/HC Set-Up

The T-Sonic OM (length, 14.5 cm; diameter, 3.6 cm; optimum flow 3.9 L/min) and PW (length, 29 cm; diameter 3.9 cm; optimum flow 9 L/min) cavitators (Treelium: Stabio, Switzerland), suitable for descaling and water sanitation, respectively, were used to assess the disinfection effectiveness of cavitation, individually and in combination with UVC. For assessing the individual disinfection efficiency of HC, the device simply replaced the UVC system in the specified set-up (Figure 1a). To explore the bacterial inactivation efficiency of HC in hybrid schemes with UVC, the cavitators were inserted before (HC-UVC) or after the UVC (UVC-HC) system (Figure 1a; technology combination junctions). Standard pressure gauges were used to monitor inlet and outlet pressure.

### 2.1.3. ClO$_2$ Generation

The Lotus mini 20 chlorine dioxide generator (Emec: Rieti, Italy) was purchased for the stable and safe production of ClO$_2$ from HCl (9% ($w/v$)) and NaClO$_2$ (7.5% ($w/v$)). Briefly, RO or borehole water was pumped into the generator at 10–20 L/min. Input water was subjected to carbon filtration to ensure removal of chlorine and organic residual. The precursor chemicals were pumped with specific proportions into the generator, with strokes externally controlled. On reaction, ClO$_2$ was generated at a maximum of 2 g/L, proportionally to the pressure and the flow rate of the circulating water. ClO$_2$ exited the generator through a back-pressure valve at the top of the chamber and was used to subsequently dose *E. coli-* and *E. faecalis*-spiked water samples at the desired concentration and contact time (C$_t$) (Figure 1b).

### 2.1.4. Eletrochlorination

For the electrochemical generation of sodium hypochlorite (in situ chlorination), the Minichlorgen 30 (Lutz-Jesco, Wedemark, Germany) was utilised. RO or borehole water was fed into the system at 5 L/h. Input water was subjected to carbon filtration to ensure removal of chlorine and organic residual. Water was softened before diluting the brine and dissolving the salt. Diluted brine entered the electrolyser at the same flow rate, allowing for chloride oxidation to chlorine (anode) and salt reduction to NaOH and H$_2$ (cathode). Released chlorine interacted with NaOH to generate NaClO with H$_2$ gas release. The produced NaClO (6 g/L) was collected in the designated tank (Figure 1c) and a dosing pump was used to treat *E. coli-* and *E. faecalis*-spiked water samples at the desired doses and C$_t$.

### 2.2. Bacterial Strains and Culture Conditions

For spiking autoclaved and/or filtered deionised (pH 7.2 ± 0.2, 20.5 ± 2.5 °C) and/or RO water (pH 7.5 ± 0.2, 20.5 ± 2.5 °C), two non-pathogenic strains were used; *Escherichia coli* O157:H7 NCTC 12900 (National Collection of Type Cultures of the Health Protection Agency (London, UK)), and *Enterococcus faecalis* NCTC 775 (ATCC 19433; TCS Biosciences: Buckingham, UK), with *E. faecalis* generally appearing more resistant to both traditional chlorination and to alternative disinfection technologies than *E. coli* [31]. Bacterial glycerol (30% *v/v*) stocks in tryptic soy broth without dextrose (TSB; Scharlab: Barcelona, Spain) were stored at −80 °C. For preparing fresh bacterial suspensions, the detailed procedure described in Text S2 was followed. The initial bacterial load of the spiked water was on average 20 × 10$^8$ cfu/mL.

### 2.3. Bacterial Inactivation Treatments

Bacterial inactivation treatments were conducted in triplicate, unless otherwise specified, with at least one replicate performed on a separate day. Samples were collected immediately after each treatment and subjected to bacteriological and/or chemical analyses.

### 2.3.1. UVC

For bench-scale evaluations, *E. coli-* and/or *E. faecalis*-spiked deionised or RO water was subjected to UVC-disinfection at 0.17–9.50 L/min and minimum exposure times (t$_{RMin}$; 0.27–15.2 min for laminar flow and 0.17–9.2 min for turbulent flow), amounting to a range of radiation doses/fluences (0.4–22 kJ/m$^2$, Supplementary Materials Table S1). The t$_{RMin}$ exposure times were calculated according to Text S3 [32]. With the proposed set-up (Figure 1a), extension of the exposure time by 3- to 5-fold required recycling of the treated water 3–5 times, simulating scenarios where running water enters a series of 3–5 UVC units. For preliminary plant-pilot studies on borehole water (Ta' Saliba borehole, Chadwick Lake origin; pH 8.5 ± 0.1, 18.5 ± 1.3 °C), the UVC device was installed on site, and its disinfection efficiency was assessed for a single exposure over the 6.5–17 L/min flow rate range (t$_{RMin}$ 9–24 s).

### 2.3.2. HC and UVC/HC

To explore the disinfection efficiency of HC when combined with UVC, the cavitator was inserted before (HC-UVC) or after the UVC device (UVC-HC) (Figure 1a). The 9.5 and 15 L/min flow rates were chosen for the delivery of *E. coli*- and *E. faecalis*-spiked water to the devices (inlet pressure; $1.7 \pm 0.6$ bar), respectively, since at these flow rates, UVC had individually exhibited the lowest $Log_{10}$ microbial inactivation. Bacterial inactivation at the specified flow rates was assessed for single, triple, and quintuple passages. The microbial inactivation efficiency of HC alone was also studied at the same flow rates and exposure times for comparison purposes. For preliminary pilot studies, the T-Sonic PW hydrocavitator was installed at the Saliba borehole. Over single exposures, groundwater entered the cavitator at 6.5–17 L/min, with an inlet pressure of $2.0 \pm 0.1$ bar and an outlet pressure of $1.0 \pm 0.1$ bar ($\Delta p = 1$ bar).

Prior to the treatment of borehole water by either UVC or HC, the pipe system was washed at full flow for at least 5 min, to ensure discharge of possible bacterial growth. After setting the desired flow, the system was allowed to equilibrate for a minimum of 5 min before treatment application and sample collection. For borehole water, bacterial inactivation was assessed on the total bacterial count (TBC) at 37 °C.

### 2.3.3. ClO$_2$ and NaClO

Following the electrochemical generation of 2 g/L ClO$_2$ and 6 g/L NaClO (continuous modes), 250–500 mL filtered RO water, spiked with 1% (*v/v*) *E. coli*- or *E. faecalis*-Ringer's suspension, was individually dosed with 0.3, 0.75, 0.9, 1.9, 3.8, 7.5 and 15 mg/L ClO$_2$ or 0.15, 0.3, 0.75, 1.1 and 2.1 mg/L NaClO. The bactericidal was allowed to act for 0 s, 30 s, 2 min, 5 min, 10 min, 30 min, 45 min, 1.5 h and 2.5 h, before addition of 0.1 M sodium thiosulphate pentahydrate (Sigma-Aldrich) for quenching the bactericidal residual and terminating the reaction [33]. Samples were then subjected to bacteriological and/or chemical analyses. Since the natural bacterial load of borehole water lacked *E. faecalis*, water samples from the Saliba borehole were spiked with *E. faecalis* and were individually dosed with the two lowest disinfectant concentrations, i.e., 0.3 and 0.75 mg/L ClO$_2$ or 0.15 and 0.3 mg/L NaClO. Treatments were allowed to proceed for the same $C_t$s as for the RO water, before reaction quenching and sampling for further analyses.

### 2.4. Colony Counting and Bacterial Inactivation

For colony counting, spread-plating was opted for the bacteriological analysis of 50–1000 μL sample volumes, pour-plating for 1.5–5 mL volumes, and filtration for sample volumes > 5 mL. Untreated spiked samples and treated spiked samples, particularly over low fluences/doses and short $C_t$s, were serially diluted in Ringer's solution before plating to ease colony enumeration. Samples were plated on TSA, before colony identity confirmation/verification by use of selective media (Text S2) [34]. Bacterial $Log_{10}$ or Ln reduction was determined using Equations (1) and (2), respectively, to facilitate direct comparisons to previously published works (Text S4).

$$Log_{10} reduction = Log_{10} \frac{N}{N_0} \tag{1}$$

$$Ln\ reduction = Ln \frac{N}{N_0} \tag{2}$$

where $N_0$ and $N$ were the bacterial-colony-forming units per mL of suspension (cfu/mL), before (influent) and after treatment (effluent), respectively. The limit of inactivation for *E. coli* and *E. faecalis* in drinking water was set at 0 colonies per 100 mL (EU directive).

### 2.5. Bactericidal Decomposition and Breakpoint Chlorination

For determining the half-lives of ClO$_2$ and NaClO in RO and/or borehole water, ClO$_2$ and residual chlorine concentrations were determined using the chlorophenol red

(Hanna instruments, Catania, Italy) and N,N-diethyl-p-phenylenediamine (DPD; Palintest, Gateshead, UK) methods, respectively, according to the manufacturer instructions, in free-from biological demand samples, over 24 h. Absorbance of the samples was recorded using the H197738 photometer (Hanna instruments, Padua, Italy). For determination of breakpoint chlorination points, DPD-1 and DPD-3 (Lovibond, Salisbury, UK) reagents were used for the colourimetric detection of free and total chlorine residual, respectively.

### 2.6. Bacterial Inactivation Kinetics

$Log_{10}$ bacterial inactivation versus fluence or time were plotted in Excel or Graphpad prism (Dotmatics, Boston, MA, USA). GinaFit [35] was employed for testing different inactivation models (Text S4). UVC-based inactivation of deionised water was best described by the Weibull + tail model, whereas the $Log_{10}$ linear function was used to fit the data derived from the UVC-inactivation of the TBC (37 °C $\pm$ 2 °C) in borehole water (Text S4.1) [36–38]. Chemical disinfection treatments were best described with a modified version of Hom's model (Text S4.2) [39–42].

### 2.7. Determination of Indicator Parameters and Chemical Analyses

Conductivity, pH, dissolved oxygen (DO), total dissolved solids (TDS), oxidation-reduction potential (ORP), and turbidity were routinely measured using a multi-meter (HI9829, Hanna instruments, Italy). Determinations of biological oxygen demand (5210-B), total alkalinity (2320-B), total hardness (2340-C), $Ca^{2+}$ hardness (2340-C), $Mg^{2+}$ hardness (2340-C), total suspended solids (2540-B/2540-D), and chlorides (4500-Cl-B) were carried out by the Water Service Corporation, Triq Hal Qormi Ħal Luqa, Malta (ISO/IEC 17025: 2017) according to the standard methods for the examination of water and wastewater [43]. Determinations of total non-ferrous metals, nitrates, nitrites, chlorides, total organic carbon (TOC), chloramines, halo-acetic acids (HAAs)/halogenated acetonitriles, THMs (mainly chloroform, bromoform, dibromochloromethane, dichlorobromomethane), polycyclic aromatic hydrocarbons (PAHs), halogenated/non-halogenated organic volatiles, chlorate and chlorite were conducted by ALS (ALS (Brno, Czech Republic) and were covered by ISO 17025:2017 accreditation. For the ALS analyses of non-chemically disinfected samples, borehole water was subjected to (a) radiative treatment at 17, 14, 10 and 6.5 Lt/min (corresponding to 9, 11, 15.5 and 24 s minimum exposure times, respectively) or (b) HC, at the same flow rates (inlet pressure 2 bars). For both technologies, samples were collected after a single exposure/passage. For chemical disinfection, borehole water was subjected to 0.3 or 0.75 mg/L $ClO_2$ or NaClO for two hours.

### 2.8. Statistical Analyses

Mean values of triplicate measurements with their standard deviations (STDEV) and/or errors (SEM) were determined using Excel. To assess the statistical strength of comparisons, *t*-tests were performed in GraphPad Prism at 95% confidence intervals ($p < 0.05$). To assess the goodness of the fitted-to-the data models, the parameters described (Text S5) were used.

### 2.9. Feasibility Assessments

For the identification of the most suitable technologies for implementation in larger scale on-site studies, a feasibility assessment tool was generated (Table 1) [44], evaluating them on (a) network integration, (b) adaptability, (c) practicality, (d) cost-effectiveness, and (e) environmental sustainability [45]. The performance of each technology was evaluated by addressing a set of questions, with marks allocated to the technology based on the provided answer (Table 1).

**Table 1.** Feasibility assessment tool with its marking scheme.

| Feasibility Areas | Questions | Answers | Assessment Scheme |
|---|---|---|---|
| 1. Implementation | 1. Can the technology be implemented given the current water characteristics? | Yes<br>No<br><br>Yes, with provisions [1] | 1 point<br>0 points<br>1–5 provisions (0.75 points)<br>6–10 provisions (0.5 points)<br>11–15 provisions (0.25 points) |
| 2. Practicality | 1. Is operation easy? | Yes<br>No<br>Yes, with provisions | 1 point<br>0 points<br>1–5 provisions (0.5 points)<br>>5 provisions (0.25 points) |
| | 2. Is installation easy? | Yes<br>No<br>Yes, with provisions | 1 point<br>0 points<br>1–5 provisions (0.5 points)<br>>5 provisions (0.25 points) |
| | 3. Is operation occupationally safe? | Yes<br>No<br>Yes, with provisions | 1 point<br>0 points<br>1–5 provisions (0.5 points)<br>>5 provisions (0.25 points) |
| | 4. Is dosing low on maintenance? | Yes<br>No | 1 point ($\geq$1 advantage)<br>0 points |
| 3. Adaptability | 1. Does the technology offer flexibility? | Yes<br>No | 1 point ($\geq$1 advantage)<br>0 points |
| | 2. Is the technology easily adaptable in a cost-effective manner? | Yes<br>No | 1 point ($\geq$1 advantage)<br>0 points |
| 4. Integration | 1. Is the technology suitable for treatment of RO/blended water? | Yes<br>Partly<br>No | 1 point<br>0.5 points (1 disadvantage)<br>0 points (>1 disadvantage) |
| | 2. Is the technology suitable for treatment of borehole water? | Yes<br>Partly<br>No | 1 point<br>0.5 points (1 disadvantage)<br>0 points (> 1 disadvantage) |
| | 3. Can the technology be combined with other technologies in hybrid schemes? | Yes<br>No<br><br>Yes, with provisions | 1 point<br>0 points<br>1–5 provisions (0.75 points)<br>6–10 provisions (0.5 points)<br>11–15 provisions (0.25 points) |
| | 4. Does the technology require specific chemical analyses techniques for monitoring? | Yes<br>No | 0 points ($\geq$1 techniques)<br>1 point |

**Table 1.** *Cont.*

| Feasibility Areas | Questions | Answers | Assessment Scheme |
|---|---|---|---|
| 5. Environment and Sustainability | 1. How does the technology rank in terms of $CO_2$ emissions? [2] | 1–3 ranks<br>4th rank | 3 (1st), 2 (2nd), 1 (3rd) points<br>0 points |
| | 2. Is the technology more energy-efficient than gas chlorination? | Yes<br>No | 1 point<br>0 points |
| | 3. Would installation of the technology at the reservoir level pose additional environmental effects? | Yes<br>No | 0 points<br>1 point |
| | 4. Would installation of the technology at the borehole level pose additional environmental effects? | Yes<br>No | 0 points<br>1 point |
| | 5. Could the technology be powered by alternative energy means? | Yes<br>No | 1 point<br>0 points |
| 6. Cost and Effect | 1. Is application of the technology for RO/blended water fit for purpose (safe and clean water for consumption)? | Fully<br>No<br>Partly | 1 point<br>0 points<br>0.5 points |
| | 2. Is application of the technology for borehole water fit for purpose (safe and clean water for consumption)? | Fully<br>No<br>Partly | 1 point<br>0 points (>2 disadvantages)<br>0.5 points (2 disadvantage) |
| | 3. In terms of microbial inactivation, how does the technology perform? | | Total points from disinfection assessment |
| | 4. In terms of by-products, is the technology likely to improve the water's taste? | Yes<br>Possibly<br>No | 1 point (1 DBP)<br>0.5 points (Few DBPs)<br>0 points ($\geq$2 DBPs) |
| | 5. How costly is application of the technology at RO level? | | Total points from cost analysis |
| | 6. How costly is application of the technology at borehole level? | | Total points from cost analysis |

Notes: [1] A table with the corresponding provisions (Supplementary Materials Table S7) is available in the supplementary section. [2] kWhs of electricity of instrumental operation for the exclusive purpose of dosing were converted to kgCO₂e (kg $CO_2$ equivalents) using the 0.391 kgCO₂e per kWh factor reported for Malta, by Carbon Footprint Ltd. (Basingstoke, UK), in their 2022 report.

Financial (Supplementary Materials Tables S2–S4 based on the assumptions of Section 2.9.1), chemical analyses (Section 3.5) and bacterial inactivation (Supplementary Materials Tables S5 and S6) outcomes were used to address the 6th feasibility area questions (Table 1) and generate Supplementary Materials Tables S7 and S8.

### 2.9.1. Financial Analyses Assumptions

Financial analyses focused on determining capital (CAPEX) and operational/maintenance (OPEX) expenditures, when implementing each technology in groundwater, desalinated water, and blended water treatments. Costs for equipment purchase, shipping, installation, commissioning, and staff training were considered for CAPEX analyses, whereas the annual costs for equipment service, part replacement, preventive maintenance, and consumables defined the OPEX analyses (Supplementary Materials Tables S2–S4).

Costs for the implementation of the individual alternative-to-gas-chlorination technology in the treatment of groundwater, RO water, and blended water required the definition of equipment sizing. For sizing decisions, water flow rates supplied in the Maltese islands by the WSC during the 2021 peak season were considered; namely, 14 $m^3$/h for groundwater abstraction from boreholes and pumping stations, 2689.6 $m^3$/h for RO water, and 3013.46 $m^3$/h for blended water supplied from the reservoirs. Generally, each instrument unit within budget could satisfy a maximum flow rate of 100 $m^3$/h. Consequently, for calculating purchasing and operational costs for the corresponding technology, we considered the product of cost per unit and the number of units required to match the 2021 peak season flow rates. Installation, commissioning, and on-site staff training costs were provided as a block-sum by one supplier. Thus, this block-sum was assumed for each of the analysed scenarios. As costs for purchasing the individual equipment parts for each technology were not always evident, the cost of parts was considered as the block-sum provided by the respective supplier. Energy costs were determined for the maximum equipment power consumption at a rate of 0.1275 €/kWh excluding VAT. The cost of quality control was taken as the cost for the continuous operational water quality monitoring currently practiced. For the different disinfection scenarios, additional monitoring costs associated with the specific disinfection by-products for each technology were considered. Finally, since the costs for equipment servicing per annum had not been detailed by all the potential suppliers, the cost of parts was considered as the block-sum provided by the respective supplier.

The labour hours required for the operation and maintenance of the equipment for each of the disinfection technologies presented in the analysis are estimated on the basis of the following assumptions: (a) for UVC 1 h per week and 4 h per month will be dedicated to visual checks, and lamp/sleeve maintenance, respectively (total 8 h per month), and (b) for chlorine dioxide and electrochlorination, 1 h per week and 4 h per month will be dedicated to visual checks, and for replenishing consumables, respectively (total 8 h per month). We have assumed that for each technology, two WSC personnel in the posts of Technician 1 (paid at 11.83 €/h) and Technician 2 (paid at 12.65 €/h) will attend to the equipment. Where more than one unit is considered per alternative-to-gas-chlorination disinfection technology, the labour cost is adjusted to the number of operable units (Supplementary Materials Tables S2–S4).

## 3. Results

### 3.1. UVC-Mediated Bacterial Inactivation

The $Log_{10}$ inactivation profiles for both microorganisms exhibited a biphasic character comprising a fast inactivation phase (up to $1.0 \pm 0.5$ $kJ/m^2$) and a slower one featuring prolonged tailing for low-to-medium irradiances and prolonged exposure (Figure 2a). The fast phase of the curves showed upward concavity (Figure 2b) suggestive of sturdier bacterial populations becoming harder to inactivate over time (for further clarifications Text S4 is provided). The upward concavity of the *E. coli* dose-response curve was 1.5-fold lower than that of *E. faecalis* ($p = 0.0211$, $n = 12$), implying exhibition of greater *E. coli* resilience over the first phase (Table 2). However, the *E. faecalis* 4D (fluence for 4 decimal reductions)

and $\text{Log}_{10}(N_{res})$ values were 1.75-fold and 2-fold higher than those for *E. coli*, respectively (Table 2), indicating that UVC was less effective at inactivating *E. faecalis* (Figure 2b, Table 2). With $R^2$ adj. close to 1, RMSE < 0.7, and RMSE almost matching the STDEV values, the chosen model fitted the data adequately (Table 2).

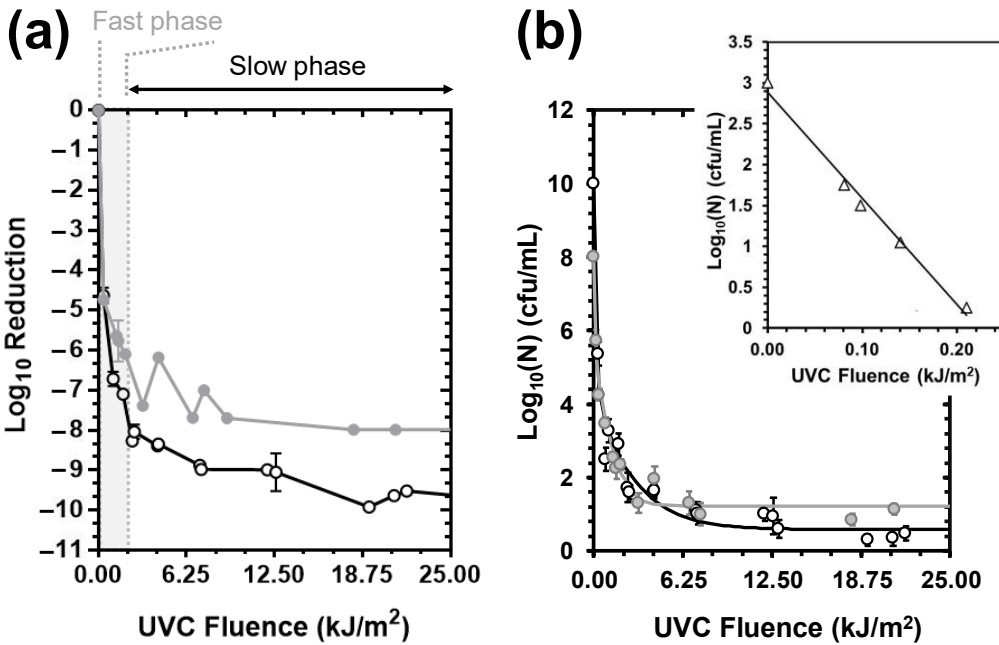

**Figure 2.** *E. coli* and *E. faecalis* inactivation in UVC-treated deionised water. (**a**) $\text{Log}_{10}$ reduction with applied fluence. Shaded area highlights the fast phase of the inactivation. (**b**) $\text{Log}_{10}(N)$ changes with applied fluence. The Weibull + tail function was fitted to the *E. coli* and *E. faecalis* data, since the fluence remained constant during contact. Inset shows the $\text{Log}_{10}(N)$ inactivation for the TBC (37 °C) (up to 0.25 kJ/m$^2$). Linear regression was fitted to the TBC (37 °C) data (single measurements). Fluences $\leq$ 18 kJ/m$^2$ correspond to flow rates in the range of 1.71–15 L/min and a minimum time of 0.17 min. Doses > 18 kJ/m$^2$ correspond to flow rates in the range of 0.083–1.7 L/min and a minimum time of 1.52 min. Bars depict the standard error (SEM) of triplicate measurements. *E. coli*; open circles-black line, *E. faecalis*; grey closed circles-solid line, and TBC (37 °C); open triangles-solid line.

**Table 2.** Derived Weibull + tail model parameters for the UVC inactivation of *E. coli*- and *E. faecalis*-spiked deionised water and of TBC (37 °C) in borehole water.

| Sample | p | $\delta$ (J/m$^2$) | 4D (kJ/m$^2$) | $\text{Log}_{10}(N_{res})$ (cfu/mL) | $R^2$ adj. | MSE | RMSE | STDEV of Residuals |
|---|---|---|---|---|---|---|---|---|
| *E. coli* | 0.22 ± 0.03 | 0.19 ± 0.24 | 0.36 ± 0.08 | 0.56 ± 0.14 | 0.9803 | 0.1202 | 0.3467 | 0.3464 |
| *E. faecalis* | 0.34 ± 0.04 (0.0211) [1] | 10.4 ± 6.19 (n.s.) [1] | 0.63 ± 0.06 (0.0188) [1] | 1.08 ± 0.12 (0.0113) [1] | 0.9879 | 0.0581 | 0.2410 | 0.2390 |
| TBC (37 °C) | - | 82.0 ± 0.02 (0.0001) [2] | n.d. [3] | - | 0.9955 | 0.0365 | 0.1910 | 0.1915 |

Notes: Parameters are defined in Supplementary Text S4. [1] *p*-values for comparisons to the parameters of *E. coli*; n.s. stands for non-significant. [2] *p*-value for comparisons to both *E. coli* and *E. faecalis* parameters. [3] n.d.; not-determined by GinaFit: Leuven, Belgium.

The fluence required to achieve the first decimal reduction of TBC (37 °C) was at least 8-fold higher (Table 2) than the fluences required to achieve the same reductions of *E. coli*- and *E. faecalis*-spiked deionised water. This was probably due to the distinct composition of the water's natural bacteriological content, with different species requiring different fluences for complete inactivation [6]. Lowering the flow rates and prolonging the exposure times progressively achieved a maximum 2.75 $\text{Log}_{10}$ reduction of TBC (37 °C) (Figure 3b).

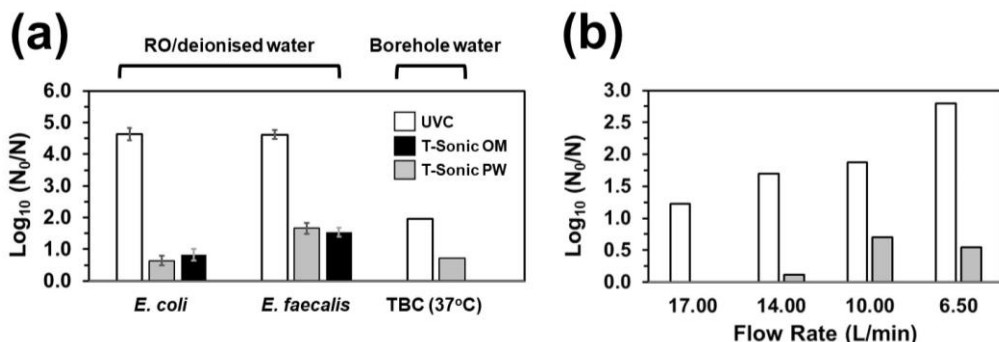

**Figure 3.** Comparative evaluation of the inactivation efficiencies of non-chemical disinfection technologies in the treatment of potable water. (**a**) Comparison of the $\text{Log}_{10}$ ($N_0/N$) values in TBC (37 °C) of borehole water to the inactivation efficiencies of UVC and HC (cavitator PW) in the treatment of *E. coli*- and *E. faecalis*-spiked RO/deionised water. Comparisons are provided for a single exposure/passage at 10–11.5 L/min flow rate. (**b**) TBC (37 °C) inactivation profiles of UVC- and HC-treated borehole water, at different flow rates. Bars denote the SEM of triplicate measurements. Borehole water disinfection data stem from single measurements.

### 3.2. HC-Mediated Bacterial Inactivation in Borehole Water

Based on the performance of the T-Sonic OM and PW cavitators (Text S6), OM was employed in subsequent UVC/HC hybrid treatments to investigate whether HC effects could enhance the achieved UVC-mediated inactivation at higher flow rates and/or short exposure times (low fluences). Due to the capacity of PW to tolerate higher flow rates, it was implemented into the pipework network of Ta'Saliba borehole to assess its decontamination efficiency. Individually, the PW achieved <1 $\text{Log}_{10}$ inactivation of TBC (37 °C) (Figure 3a), a performance matching the disinfection of *E. coli*-spiked RO/deionised water (Supplementary Materials Figure S3). Interestingly, HC-PW effects on borehole water were maximised at a 10 L/min flow rate (Figure 3b), suggesting an optimal reduction of the local pressure relative to the vapour fluid pressure and instigation of effective cavitation inception [46].

### 3.3. UVC/HC Hybrid Treatment-Mediated Inactivation

For *E. coli*-spiked deionised water, hybrid UVC/HC effects were 1.3-fold stronger than UVC alone, when HC succeeded UVC and only for a single passage (Figure 4a top-left). However, equally significant HC/UVC effects were evident over prolonged UVC-exposure times when HC preceded the UVC treatment (Figure 4a top-left). Differences between hybrid treatments were insignificant. T-test-based comparisons (Text S7) between the *E. coli* $\text{Log}_{10}$ ($N_0/N$) achieved by the UVC/HC hybrid treatments and the summation of the $\text{Log}_{10}$ ($N_0/N$) values stemming from the use of the individual technologies revealed no statistically significant differences ($p > 0.16$, $n = 3$), suggesting that the $\text{Log}_{10}$ ($N_0/N$) changes of the hybrid-treatment more likely arose from additive effects of the individual technologies (Figure 4b top-right).

Similarly, *E. faecalis* inactivation in deionised water by the UVC-HC hybrid treatments was at least 1.4-fold higher than the individual UVC treatment, regardless of exposure length and cavitator type ($p = 0.0009$, $n = 3$) (Figure 4a bottom-left). T-test-based comparisons between the *E. faecalis* $\text{Log}_{10}$ ($N_0/N$) achieved by the UVC/HC hybrid treatments and the summation of the $\text{Log}_{10}$ ($N_0/N$) values stemming from the use of the individual technologies revealed no statistically significant differences ($p > 0.16$, $n = 3$) for first passage treatments (Figure 4b bottom-left). However, on subsequent passages (5th; Figure 4b bottom-left), summation of the $\text{Log}_{10}$ ($N_0/N$) values achieved by the individual technologies failed to match the values stemming from the hybrid treatments, implying the emergence of synergism between the two technologies.

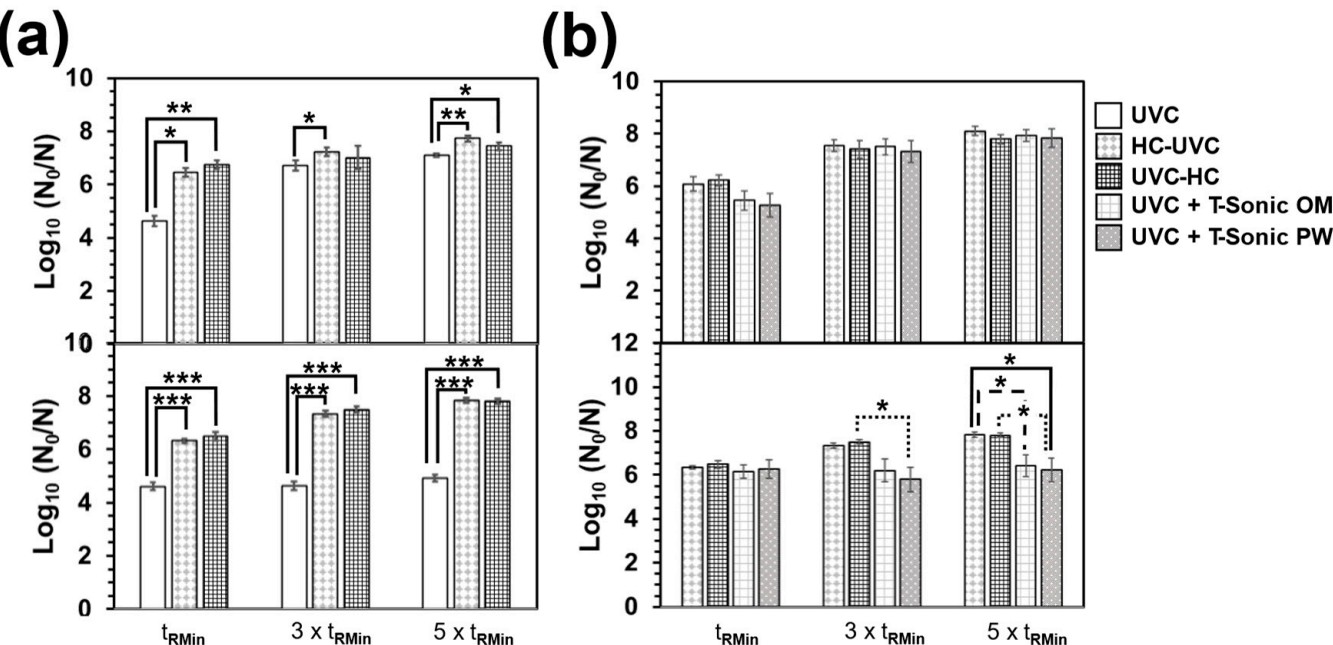

**Figure 4.** Bacterial $Log_{10}$ ($N_0/N$) changes by UVC and HC/UVC over exposure time. (**a**) Top: UVC and UVC/HC treatments of *E. coli*-spiked deionised water at 9.5 L/min. The applied UVC doses for minimal exposure times of 0.27 ($t_{RMin}$), 0.81 ($3 \times t_{RMin}$), and 1.35 ($5 \times t_{RMin}$) min, were 0.35, 1.05 and 1.75 $kJ/m^2$, respectively. Bottom: UVC and UVC/HC treatments of *E. faecalis*-spiked deionised water at 15 L/min. The applied UVC doses for 0.17 ($t_{RMin}$), 0.51 ($3 \times t_{RMin}$), and 0.85 ($5 \times t_{RMin}$) min, were 0.25, 0.78 and 1.34 $kJ/m^2$, respectively. Bars represent the STDEV of triplicate measurements. Differences between HC/UVC combinations were insignificant. (**b**) Top: comparison of *E. coli* inactivation achieved by hybrid UVC/HC treatments and the summation of the effects of the constituent technologies at 9.5 L/min. Bottom: comparison of *E. faecalis* inactivation achieved by hybrid UVC/HC treatments and the summation of the effects of the constituent technologies at 15 L/min. Exposure times are as in (**a**). Hyphenation indicates the order in the hybrid treatment, i.e., HC before UVC (HC-UVC). For the 9.5 L/min flow, HC exposures were 0.016, 0.0048 and 0.08 min, for single, triple, and quintuple passages, respectively. For the 15 L/min flow HC exposures were 0.023, 0.069 and 0.115 min, for single, triple, and quintuple passages, respectively. Bars represent the SEM of triplicate measurements. Significance at 95% confidence intervals (*n* = 3) is denoted as: $p \leq 0.05$ (*), $p = 0.001$ to 0.01 (**), and $p = 0.001$ to 0.0001 (***).

Prolonged exposure of *E. faecalis*-spiked deionised water to UVC/HC hybrid-treatments using the OM cavitator was associated with an increase in ORP (from third exposure onward), and oxygen saturation and % DO by the fifth exposure (Figure 5).

For cavitator PW-treated tap water, there was a nearly 9% decrease in the concentration of TDS, which coincided with a similar percentage reduction in conductivity by the fifth passage (Figure 6), consistent with the elimination of organic/inorganic matter. In addition, there was at least an 8% reduction in pH and a maximum of 39% increase in ORP (Figure 6), indicative of mild acidification of water and the generation of reactive oxygen species.

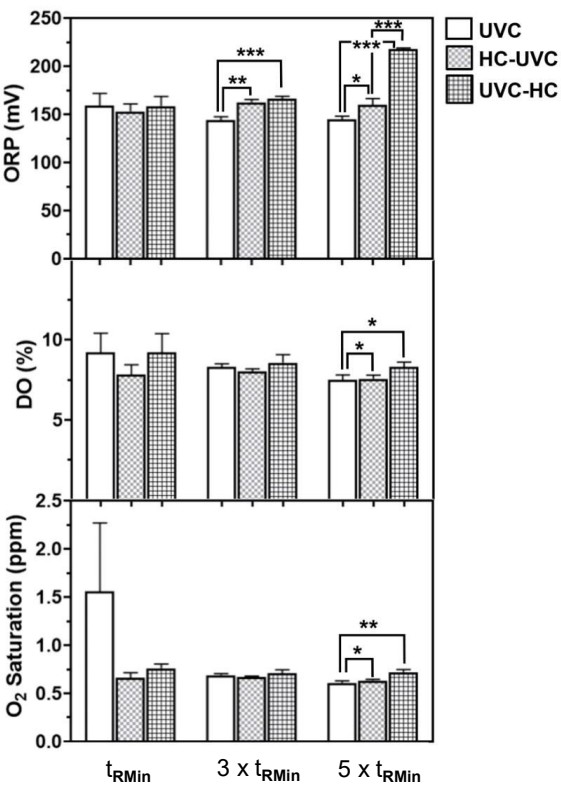

**Figure 5.** Mean ORP, DO, and $O_2$ saturation measurements for *E. faecalis*-spiked deionised water (pH 7.2 ± 0.2, 20.5 ± 2.5 °C) for UVC and HC/UVC hybrid treatments (using the PO cavitator) at 15.0 L/min over different contact times. The applied UVC doses for minimal exposure times of 0.17 ($t_{RMin}$), 0.51 (3 × $t_{RMin}$), and 0.85 (5 × $t_{RMin}$) min, were 0.25, 0.78 and 1.34 kJ/$m^2$, respectively. HC exposures were 0.023, 0.069 and 0.115 min, for single, triple, and quintuple passages, respectively. Bars represent the standard deviation of triplicate measurements. Significance at 95% confidence intervals (*n* = 3) is denoted as: $p \leq 0.05$ (*), $p = 0.001$ to 0.01 (**), and $p = 0.001$ to 0.0001 (***), for the differences between HC/UVC combinations or for the differences between HC/UVC combinations and UVC alone. Absence of stars denotes no significance.

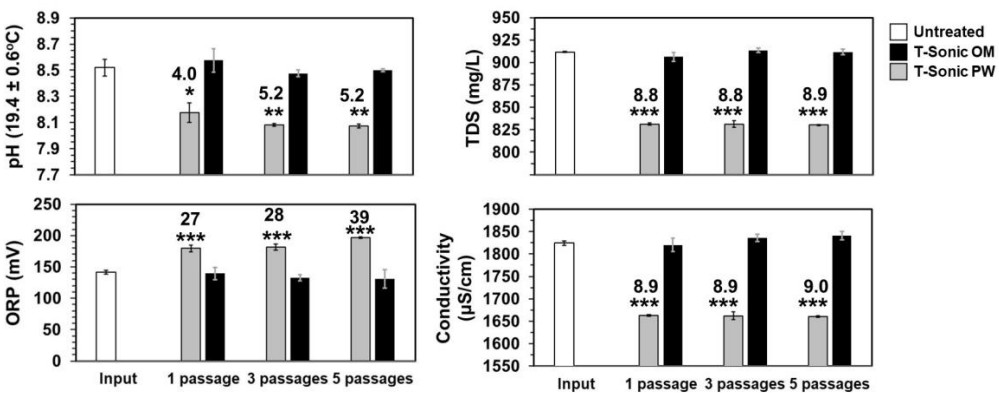

**Figure 6.** Mean pH, ORP, TDS, and conductivity measurements of tap water following treatments with T-Sonic OM (9.5 L/min) and T-Sonic PW cavitators (15 L/min) over different passages. For the 9.5 L/min flow, HC exposures were 0.016, 0.0048 and 0.08 min, for single, triple, and quintuple passages, respectively. For the 15 L/min flow HC exposures were 0.023, 0.069 and 0.115 min, for single, triple, and quintuple passages, respectively. Bars represent the standard deviation of triplicate measurements. Asterisks denote level of significance for 95% confidence for comparisons to the input: $p = 0.02$–0.05 (*), $p = 0.001$–0.01 (**), and $p \leq 0.001$ (***) for *n* = 3. Numbers above asterisks denote the percentage change relative to the input.

*3.4. Chemical Disinfection*

3.4.1. Decomposition Half-Lives and Chemical Demand

Before assessing the decontamination potential of the chemical disinfection technologies, decomposition half-lives of $ClO_2$ and $NaClO$ in RO (pH 7.5) and borehole (pH 8.4) waters at different concentrations were determined (Supplementary Materials Figures S4 and S5). In harmony with published works [47–52], alkalinity encouraged $ClO_2$ decomposition (Text S8, Figure S4b), yielding half-lives 19-fold and 36-fold shorter than at pH 7.5 for 0.75 and 0.3 mg/L $ClO_2$, respectively ($p < 0.0001$, $n = 21$) (Supplementary Materials Table S9). In contrast, $NaClO$ generated in RO (Supplementary Materials Figure S5a) and/or borehole water (Figure S5b) exhibited half-lives in the 2.7–4.2 hr range (Table S9). $NaClO$ decomposition exhibited no pH dependency ($p > 0.07$, $n = 15$) (Supplementary Materials Figure S5b).

Whereas RO and borehole waters posed minimal chemical demand for $ClO_2$ (Text S9 [53–56], Supplementary Materials Figure S6), they required $Cl_2$ and $NaClO$ dosing in the range of 0.6–0.9 mg/L for breakpoint chlorination (Supplementary Materials Figure S7), with borehole water generally demanding higher doses (Supplementary Materials Figure S7a). Thus, in terms of stability and chemical demand, $ClO_2$- and $NaClO$-based treatments appeared more suited for the disinfection of borehole and RO waters, respectively.

3.4.2. Decontamination Efficiency

Chemical decontamination kinetics exhibited biphasic characteristics, with a rapid phase completed within two minutes, and tailing (m < 1) (Figures 7–9). For *E. faecalis*-spiked RO water decontamination, the higher the dose, the shorter the exposure time required for 4 $Log_{10}$ reduction (Figure 7, Table 3). For the $ClO_2$ doses tested on *E. faecalis*-spiked RO water, 4 $Log_{10}$ exposure times were 1.5-fold different ($p < 0.0001$; Table 3). Similarly, for *E. coli*-spiked RO water, $ClO_2$ doses exhibited 1.6-fold different exposure times ($p < 0.0001$; Figure 8, Table 3), with the 0.75 mg/L dose achieving better effects. Both $ClO_2$ doses were at least 1.4-fold more effective in the treatment of *E. coli*-spiked than *E. faecalis*-spiked RO water ($p = 0.0003–0.0015$) (Table 3).

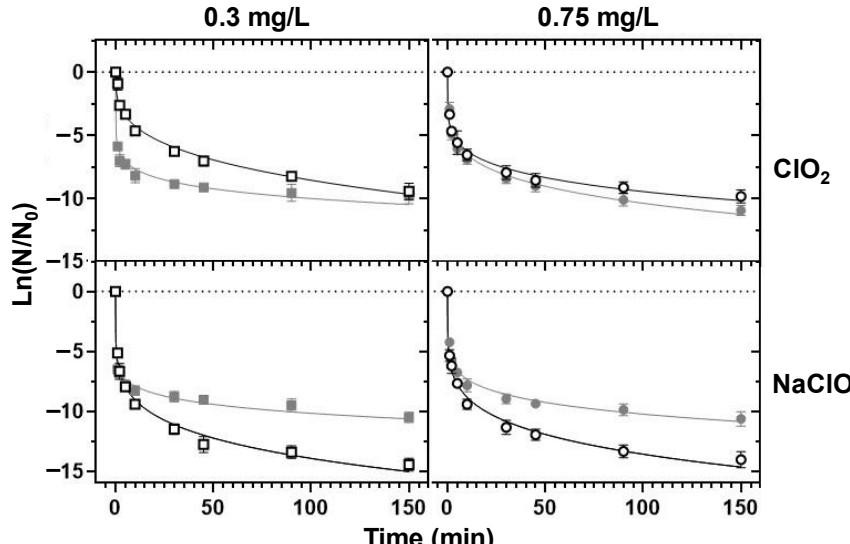

**Figure 7.** Inactivation profiles of *E. faecalis* following $ClO_2$ and $NaClO$ treatments of potable water. Top: Ln (N/N0) *E. faecalis* inactivation with time for two $ClO_2$ doses in RO (open symbols, black line) and borehole (closed symbols, grey line) waters. Bottom: Ln (N/$N_0$) *E. faecalis* inactivation with time for two $NaClO$ doses in RO and borehole waters. The modified Hom's model was fitted to the data. Bars denote the SEM of triplicate measurements.

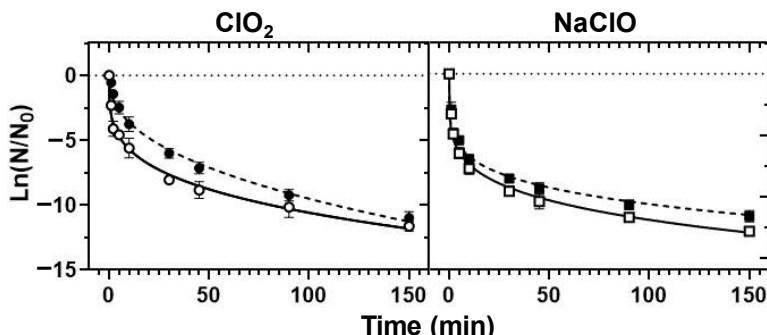

**Figure 8.** Inactivation profiles of *E. coli* following $ClO_2$ and NaClO treatments of RO water. Left: Ln $(N/N_0)$ *E. coli* inactivation with time for 0.3 (closed symbols, dashed line) and 0.75 mg/L (open symbols, black line) $ClO_2$. Right: Ln $(N/N_0)$ *E. coli* inactivation with time for 0.3 (closed symbols) and 0.75 mg/L (open symbols) NaClO. The modified Hom's model was fitted to the data. Bars denote the SEM of triplicate measurements.

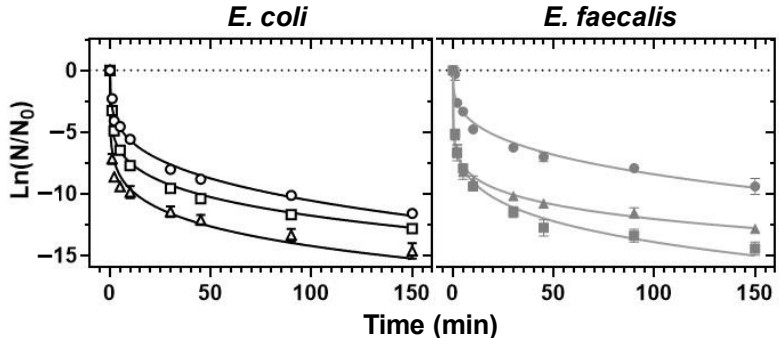

**Figure 9.** Comparative evaluation of the inactivation profiles of *E. coli* and *E. faecalis* following chemical disinfection treatments of potable water. Left: Ln $(N/N_0)$ *E. coli* inactivation with time for 1.1 mg/L total residual chlorine (open triangles), 0.75 mg/L NaClO (open squares) and 0.75 mg/L $ClO_2$ (open circles). Right: Ln $(N/N_0)$ *E. faecalis* inactivation with time for total residual chlorine (closed triangles), NaClO (closed squares) and $ClO_2$ (closed circles). The modified Hom's model was fitted to the data. Bars denote the SEM of triplicate measurements.

In the case of *E. faecalis*-spiked borehole water, the $ClO_2$ doses required similar exposure times for 4 $Log_{10}$ inactivation ($p = 0.4307$) (Figure 7, Table 3). For 0.75 mg/L $ClO_2$, the concentration exponent was > 1, whilst time exponents were similar for both doses (Table 3), indicating a higher dependency of the inactivation rate on disinfectant concentration rather than time. Overall, $ClO_2$-mediated *E. faecalis* inactivation in borehole water was at least 1.6-fold more effective than in RO (with exposure times 1.6–2.6-fold different; $p < 0.0011$; Table 3, Figure 7).

In terms of inactivation kinetics in RO water, both NaClO doses achieved comparable effects in controlling *E. faecalis* ($p = 0.1405$) (Figure 7 right, Table 3), but for *E. coli* the higher dose achieved a faster inactivation ($p < 0.0001$) (Figure 8 left, Table 3). Additionally, for *E. faecalis*-spiked borehole water treatments, both doses achieved the same inactivation at similar exposure times ($p = 0.2316$) (Figure 7, Table 3). Regardless of dose, NaClO appeared more effective at inactivating *E. faecalis* in RO than borehole water ($p < 0.0001$), with the 0.3 mg/L dose achieving a 4 $Log_{10}$ reduction nearly 4-fold faster ($p < 0.0001$) in RO (Figure 7, Table 3). Overall, NaClO was more effective at inactivating *E. faecalis* than *E. coli* (shorter exposure times; $p < 0.0001$) in RO, regardless of dose (Table 3).

Treatment of *E. coli*-spiked RO water with 0.75 mg/L $ClO_2$ was weaker than gas chlorination (1.1 mg/L total residual chlorine) of *E. coli*-spiked blended water (nearly 10-fold longer $ClO_2$ exposure; $p < 0.0001$) (Figure 9 left).

**Table 3.** Kinetic and statistic parameters determined from the modified Hom's model fitting to the chemical disinfection data.

| Water Source | Treatment | Dose (mg/L) | Organism | $k_{decay}$ (Min$^{-1}$) | $k_{disinfectant}$ (mg$^{-n}$L$^n$ Min$^{-m}$) | Exposure Time for 4 Log$_{10}$ Reduction (Min) | m | n | R$^2$ | Sy.X [1] | STDEV [1] | Degrees of Freedom |
|---|---|---|---|---|---|---|---|---|---|---|---|---|
| Blended | Cl$_2$ | 1.1 | *E. coli* | 0.0001 | 6.76 ± 0.06 [2] | 5.99 ± 0.39 [2] | 0.146 | 0.512 | 0.9858 | 0.4945 | 0.4853 | 27 |
| | | 1.1 | *E. faecalis* | 0.0001 | 5.86 ± 0.06 | 15.84 ± 1.17 | 0.146 | 0.512 | 0.9837 | 0.4825 | 0.4735 | 27 |
| RO | ClO$_2$ | 0.3 | *E. coli* | 0.001 | 3.44 ± 0.06 | 92.77 ± 3.74 | 0.420 | 0.761 | 0.9830 | 0.4947 | 0.4855 | 27 |
| | | 0.3 | *E. faecalis* | 0.001 | 2.47 ± 0.05 | 128.6 ± 8.47 | 0.300 | 0.117 | 0.9699 | 0.5462 | 0.5360 | 27 |
| | | 0.75 | *E. coli* | 0.0016 | 3.53 ± 0.06 | 58.21 ± 3.51 | 0.270 | 0.958 | 0.9786 | 0.5387 | 0.5286 | 27 |
| | | 0.75 | *E. faecalis* | 0.0016 | 4.85 ± 0.09 | 87.56 ± 8.00 | 0.180 | 0.559 | 0.9674 | 0.5521 | 0.5418 | 27 |
| | NaClO | 0.3 | *E. coli* | 0.0026 | 6.31 ± 0.10 | 48.23 ± 3.91 | 0.197 | 0.319 | 0.9734 | 0.5959 | 0.5848 | 27 |
| | | 0.3 | *E. faecalis* | 0.0026 | 7.98 ± 0.09 | 12.48 ± 0.79 | 0.187 | 0.319 | 0.9828 | 0.5700 | 0.5593 | 27 |
| | | 0.75 | *E. coli* | 0.0019 | 6.12 ± 0.08 | 28.85 ± 1.95 | 0.207 | 0.993 | 0.9794 | 0.5809 | 0.5700 | 27 |
| | | 0.75 | *E. faecalis* | 0.0019 | 8.19 ± 0.10 | 10.90 ± 0.70 | 0.187 | 1.13 | 0.9822 | 0.5969 | 0.5857 | 27 |
| Borehole | ClO$_2$ | 0.3 | *E. faecalis* | 0.04 | 16.8 ± 0.23 | 49.66 ± 6.07 | 0.115 | 0.856 | 0.966 | 0.5393 | 0.5292 | 27 |
| | | 0.75 | *E. faecalis* | 0.0403 | 5.89 ± 0.09 | 55.67 ± 4.52 | 0.200 | 1.17 | 0.9704 | 0.5885 | 0.5775 | 27 |
| | NaClO | 0.3 | *E. faecalis* | 0.0019 | 6.09 ± 0.08 | 47.94 ± 4.65 | 0.145 | 0.115 | 0.9729 | 0.5333 | 0.5233 | 27 |
| | | 0.75 | *E. faecalis* | 0.0016 | 7.30 ± 0.08 | 40.34 ± 4.22 | 0.110 | 0.600 | 0.9767 | 0.4529 | 0.4444 | 27 |

Notes: Parameters are defined in Supplementary Text S4. [1] Sy.X values were only slightly higher than the STDEV of the residuals, suggesting no significant model overfitting (Text S5). [2] Values are provided with their standard errors.

Also, gas chlorination was nearly 5-fold faster in achieving a 4-Log$_{10}$ *E. coli* inactivation than the NaClO treatment ($p < 0.0001$; Table 3). Thus, in terms of their *E. coli* inactivation efficiency, technologies were ranked as: $Cl_2 > NaClO > ClO_2$.

In contrast, for the *E. faecalis* inactivation in RO water, the NaClO treatment was 1.5-fold more effective than gas chlorination (shorter exposure time ($p = 0.0007$)) in reaching a 4 Log$_{10}$, whereas the 0.75 mg/L $ClO_2$ treatment appeared 5.5-fold less effective than chlorination (coefficient $n > 1$ and 5.5-fold longer exposure time ($p = 0.0387$)) (Figure 9 right, Table 3). Therefore, technologies were ranked as $NaClO > Cl_2 > ClO_2$ in terms of their *E. faecalis* inactivation efficiency.

### 3.5. Chemical Analyses

Both non-chemical disinfection technologies did not drastically alter the chemical content of the borehole water, with most recorded values below the parametric maxima set by the EU, suggesting that both methods were unlikely to render the potable water unsafe for consumption (Table 4). Interestingly, for both technologies there was a moderate reduction in the concentration of nitrates at high flow rates (Table 4). For HC, a reduction in the chloride content was apparent over the 6.5–10 L/min flow rates, whereas an apparent reduction in Ca$^{2+}$ hardness was evident over the 10–14 L/min flow rate range (Table 4). However, since single samples were subjected to the ALS analysis (Section 2.7) we cannot statistically validate these observations.

Chemical analyses of borehole water dosed with 0.3 and 0.75 mg/L of either $ClO_2$ or NaClO for a maximum of 2 hrs revealed parametric values below the guidance values of the EU directive for the quality of potable water (Table 5). Two trends emerged from the analysis; (1) the higher the $ClO_2$ dose, the greater the risk in exceeding the chlorate and chlorite guidance value, and (2) the higher the NaClO dose, the greater the risk in generating most of the toxic disinfection by-products also associated with gas chlorination (i.e., HAAs, THMs, chloramines and chloroacetonitriles) (Table 5; Text S10 [17,22,51,57–59]). Chloramines were present above the quantification limit in the water sample treated with 0.75 mg/L NaClO (Table 5).

### 3.6. Cost Analyses

Electrochlorination (in situ NaClO) of RO and blended waters ($\pm$ post tunnel commissioning (PTC)) would require the highest capital investment followed, by UVC. Although $ClO_2$ generation had the lowest CAPEX costs, it also had the highest OPEX costs, whereas UVC with its minimal operational and maintenance requirements was the most cost-efficient per m$^3$ (Figure 10).

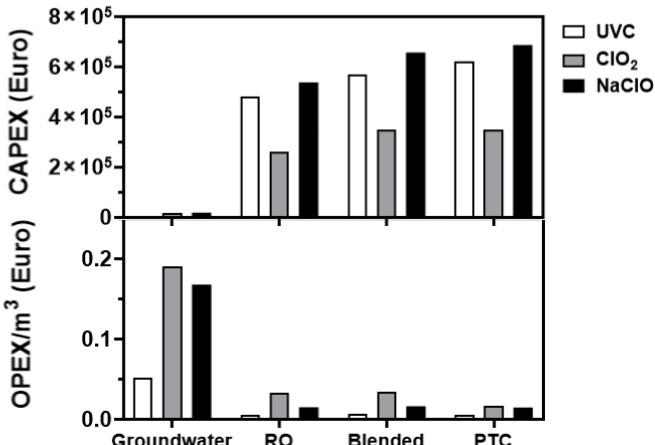

**Figure 10.** Predicted CAPEX and OPEX costs for implementation of alternative-to-gas-chlorination technologies in the treatment of Maltese potable tap water. Post tunnelling commissioning (PTC) charges were added to each technology for its application at the reservoir level (projected mean flow rate, 3401.79 m$^3$/h).

**Table 4.** Representative parametric values from the chemical analysis of borehole water samples disinfected with UVC or HC.

| | | Treatments | | | | | | | | |
|---|---|---|---|---|---|---|---|---|---|---|
| | **Eu Directive Value** | **Untreated Source** | **UVC1 (17 L/min)** | **UVC2 (14 L/min)** | **UVC3 (10 L/min)** | **UVC4 (6.5 L/min)** | **HC1 [1] (17 L/min)** | **HC2 [1] (14 L/min)** | **HC3 [1] (10 L/min)** | **HC4 [1] (6.5 L/min)** |
| Conductivity (mS/m) | - | $414 \pm 41$ | $414 \pm 41$ | $419 \pm 42$ | $412 \pm 41$ | $418 \pm 42$ | $418 \pm 42$ | $415 \pm 42$ | $412 \pm 41$ | $410 \pm 41$ |
| TOC [2] | no abnormal change | $0.81 \pm 0.16$ | $0.98 \pm 0.20$ | $1.08 \pm 0.22$ | $0.62 \pm 0.12$ | $0.72 \pm 0.14$ | $0.62 \pm 0.12$ | $1.01 \pm 0.20$ | $0.90 \pm 0.18$ | $1.72 \pm 0.34$ |
| $CaCO_3$ hardness [2] | - | 630 | 678 | 661 | 655 | 660 | 628 | 603 | 587 | 627 |
| $Ca^{2+}$ hardness [4] | - | 3.27 | 3.46 | 3.29 | 3.33 | 3.46 | 3.23 | 3.15 | 3.10 | 3.25 |
| Nitrates [2] | 50 | $49.2 \pm 7.4$ | $46.9 \pm 7.0$ | $47.3 \pm 7.1$ | $47.3 \pm 7.1$ | $47.5 \pm 7.1$ | $44.8 \pm 6.7$ | $44.7 \pm 6.7$ | $47.2 \pm 7.1$ | $46.7 \pm 7.0$ |
| Nitrites [2] | 0.5 | <0.30 | n.m. [3] | n.m. [3] | n.m. [3] | n.m. [3] | n.m. [3] | n.m. [3] | n.m. [3] | n.m. [3] |
| Calcium [2] | - | $131 \pm 13$ | $139 \pm 14$ | $132 \pm 13$ | $134 \pm 13$ | $139 \pm 14$ | $129 \pm 13$ | $126 \pm 13$ | $124 \pm 12$ | $130 \pm 13$ |
| Magnesium [2] | - | $73.6 \pm 7.4$ | $80.7 \pm 8.1$ | $80.8 \pm 8.1$ | $78.3 \pm 7.8$ | $76.2 \pm 7.6$ | $74.2 \pm 7.4$ | $69.9 \pm 7.0$ | $67.2 \pm 6.7$ | $73.5 \pm 7.4$ |
| Chloride [2] | 250 | $1220 \pm 183$ | n.m. [3] | n.m. [3] | n.m. [3] | n.m. [3] | $1260 \pm 189$ | $1270 \pm 191$ | $1150 \pm 173$ | $1140 \pm 171$ |

Notes: [1] For HC, inlet pressure was maintained at 2 bars. [2] Values are presented in mg/L (ppm); values are provided with their uncertainties in measurements (ALS Czech). [3] n.m. stands for not measured. [4] in mmol/L.

**Table 5.** Representative parametric values from the chemical analysis of borehole water samples disinfected with $ClO_2$ or NaClO (in situ $Cl_2$).

| Chemical | EU Directive Value | Untreated Source | Treatments | | | |
|---|---|---|---|---|---|---|
| | | | $ClO_2$ (0.3 ppm) | $ClO_2$ (0.75 ppm) | In Situ NaClO (0.3 ppm) | In Situ NaClO (0.75 ppm) |
| Sum of chlorate and chlorite (mg/L) | ≤0.25 natively; ≤0.7 when $ClO_2$ applied | <0.13 | 0.26 | 0.62 | <0.13 | <0.13 |
| Sum of 4 THMs (µg/L) | 100 | 0.76 | 0.79 | 1.28 | 8.98 | 17.8 |
| Dichloroacetonitrile | - | <0.10 | <0.10 | <0.10 | <0.10 | 0.20 |
| Dibromoacetonitrile | - | <0.10 | <0.10 | <0.10 | 0.50 | 0.40 |
| Sum of 9 HAAs (µg/L) | - | <20 | <20 | <20 | <20 | 42.2 |
| Sum of chloroacetic acids | - | <20 | <20 | <20 | <20 | 29.1 |
| Sum of 5 HAAs (µg/L) | 60 | <20 | <20 | <20 | <20 | 34.7 |
| Chloramines | - | <0.02 | <0.02 | <0.02 | <0.02 | 0.07 |
| Chlorate (µg/L) | 250 | <80 | 259 ± 52 | 615 ± 123 | <80 | 94 ± 19 |
| Nitrite (mg/L) | 0.50 | <0.30 | <0.30 | <0.30 | <0.30 | <0.30 |
| Nitrate (mg/L) | 50 | 45.2 ± 6.8 | 45.8 ± 6.9 | 45.6 ± 6.8 | 45.6 ± 6.8 | 45.7 ± 6.9 |
| Antimony (µg/L) | 10 | <1.0 | <1.0 | <1.0 | <1.0 | <1.0 |
| Arsenic (µg/L) | 10 | n.m. [1] | n.m. [1] | n.m. [1] | n.m. [1] | n.m. [1] |
| Boron (mg/L) | 1.5 | 0.22 ± 0.02 | 0.19 ± 0.02 | 0.19 ± 0.02 | 0.20 ± 0.02 | 0.21 ± 0.02 |
| Cadmium (µg/L) | 5.0 | <0.20 | <0.20 | <0.20 | <0.20 | <0.20 |
| Calcium (mg/L) | - | 124 ± 12 | 125 ± 13 | 127 ± 13 | 129 ± 13 | 126 ± 13 |
| Chromium (µg/L) | 25 | <1.0 | <1.0 | <1.0 | <1.0 | <1.0 |
| Copper (mg/L) | 2.0 | <0.005 | <0.005 | <0.005 | <0.005 | <0.005 |
| Iron (µg/L) | - | 35.9 ± 3.6 | 46.1 ± 4.6 | 46.3 ± 4.6 | 45.8 ± 4.6 | 47 ± 4.7 |
| Lead (µg/L) | 5.0 | 5.9 ± 0.6 | 5.6 ± 0.6 | 5.8 ± 0.6 | 6.0 ± 0.6 | 6.0 ± 0.6 |
| Manganese (µg/L) | 50 | 2.78 ± 0.3 | 2.98 ± 0.3 | 3.45 ± 0.4 | 3.05 ± 0.3 | 3.06 ± 0.3 |
| Magnesium (mg/L) | - | 66.9 ± 6.7 | 67.6 ± 6.8 | 67.6 ± 6.8 | 68.8 ± 6.9 | 69.2 ± 6.9 |
| Mercury (µg/L) | 1.0 | n.m. [1] | n.m. [1] | n.m. [1] | n.m. [1] | n.m. [1] |
| Nickel (µg/L) | 20 | 8.6 ± 0.9 | 10.7 ± 1.1 | 11.0 ± 1.1 | 9.9 ± 1.0 | 10.2 ± 1.0 |
| Selenium (µg/L) | 20 | n.m. [1] | n.m. [1] | n.m. [1] | n.m. [1] | n.m. [1] |
| Sodium (mg/Lt) | 200 | 503 ± 50 | 523 ± 52 | 510 ± 51 | 520 ± 52 | 511 ± 51 |

Notes: [1] n.m. stands for not measured. Values are provided with their uncertainties in measurements (ALS: Brno, Czech).

However, UVC-based disinfection offers no decontamination residual to safeguard the treated water from bacterial regrowth and potential biofilm formation within the distribution network. Consequently, employing the technology in the treatment of blended water (reservoir level) would be pointless for maintaining the water's safety, unless combined with an additional technology.

### 3.7. Feasibility Studies

With inputs from cost analyses (Supplementary Materials Tables S2–S4, Figure 10), bacterial decontamination efficiency (Figures 7–9, Table 3, Supplementary Materials Tables S5 and S6), DBP formation propensity (Tables 4 and 5), and potential of $CO_2$ emissions on electricity consumption (Supplementary Materials Table S8), we applied the feasibility assessment tool (Table 1) to yield Table 6. Electrochlorination ranked as the best alternative to current standard practices, followed by $ClO_2$ and UVC (Table 6).

**Table 6.** Feasibility assessment of the alternative-to-gas-chlorination disinfection technologies under study.

| Question | Implementation | Practicality | | | | Adaptability | | Integration | | | | Environment and Sustainability | | | | | | | Cost and Effect | | | | |
|---|---|---|---|---|---|---|---|---|---|---|---|---|---|---|---|---|---|---|---|---|---|---|---|
| | 1.1 | 2.1 | 2.2 | 2.3 | 2.4 | 3.1 | 3.2 | 4.1 | 4.2 | 4.3 | 4.4 | 5.1 [4] | 5.2 [5] | 5.3 [6] | 5.4 [7] | 5.5 [8] | 6.1 [9] | 6.2 | 6.3 [10] | 6.4 | 6.5 [12] | 6.6 [13] | Total |
| UVC | 0.5 (10) [1] | 1 | 0.5 (3) | 0.5 (1) | 1 | 1 | 0 | 0.5 (1) | 0 | 0.75 (3) | 1 | 2 | 0 | 0 | 0 | 1 | 0.5 | 0 (2) | 17 | 1 | 2 | 1 | 31.25 |
| ClO$_2$ | 0.5 (10) | 0.5 (1) | 0.5 (4) | 0.5 (3) | 1 | 1 | 0 | 1 | 0.5 [2] | 0.75 (3) | 0 [3] | 1 | 0 | 1 | 0 | 1 | 1 | 0.5 (1) | 20 | 1 | 0 | 2 | 33.75 |
| NaClO | 0.25 (11) | 0.5 (1) | 0.5 (5) | 0.5 (3) | 1 | 1 | 0 | 1 | 0 | 0.75 (3) | 1 | 0 | 0.5 | 1 | 0 | 1 | 0.5 | 0 (2) | 26 | 0.5 [11] | 1 | 0 | 37.00 |

Notes: [1] Numbers in parentheses denote provisions for implementation (Supplementary Materials Table S7). [2] ClO$_2$ score is higher since integration at borehole level poses fewer spatial and environmental challenges. [3] No marks were allocated, since specific analyses need to be implemented for monitoring ClO$_2$ levels. [4] In terms of CO$_2$ emissions (Supplementary Materials Table S8), UVC had the second lowest footprint (2 points), followed by ClO$_2$ (1 point) and NaClO (0 points). [5] NaClO outperformed standard chlorination only in the treatment of *E. faecalis*-spiked RO water and was allocated 0.5 points. [6] Industrial scale UVC installations have spatial requirements which would challenge integration of the technology at the reservoir level. [7] Installations at the borehole level will pose further challenges and impact negatively on the environment and wildlife. [8] All technologies could be potentially supported with solar power. [9] For RO/blended water treatments, NaClO needs to be applied before lime addition, whereas UVC imposes spatial restrictions. [10] Total marks from assessment of inactivation efficiencies. UVC (Table S5) and ClO$_2$/NaClO (Supplementary Materials Table S6). [11] NaClO is associated with the generation of more DBPs. [12] At the borehole level ClO$_2$ had the highest OPEX (0 points), followed by NaClO (1 point) and UVC (2 points) (Supplementary Materials Table S2). [13] At the RO/blended water levels NaClO had the highest CAPEX (0 points), followed by UVC (1 point) and ClO$_2$ (2 points) (Supplementary Materials Tables S3 and S4).

## 4. Discussion

### 4.1. Non-Chemical Disinfection

In agreement with previous works reporting $\geq 7$ $Log_{10}$, UVC-mediated *E. coli* inactivation [60], the UVC system of this study eliminated the $\sim 10^9$ cfu/mL *E. coli* NCTC 12900 load of the spiked water almost completely. Neither lag phases nor shoulders were present in the *E. coli* UVC-inactivation profiles, due to the absence of organic matter in the deionised water and of glucose and/or dextrose in the inoculum growing medium [61]. However, those profiles exhibited upward concavity for 0.05–0.2 kJ/m$^2$ fluences, reportedly responsible for the induction of aggregation effects and the emergence of tailing phenomena [62]. Indeed, a prolonged tail was evident in the profile for fluences $\geq 8$ kJ/m$^2$ (Figure 2). Fitting the Weibull + tail function to the *E. coli* data revealed a $Log_{10}(N_{res})$ population, indicative of larger *E. coli* flocs shielding the remaining cells from experiencing the full UVC-inactivation effect (Table 2). A 4 $Log_{10}$ reduction was attainable at $0.36 \pm 0.08$ kJ/m$^2$.

For *E. faecalis* NCTC 775-spiked deionised water a maximum 6.9 $Log_{10}$ UVC-inactivation was achieved. The *E. faecalis* UVC-inactivation profile exhibited an upward concavity with tailing (Figure 2), in harmony with the study of Moreno-Andrés et al. [63]. Our observations were also consistent with studies reporting 3 $Log_{10}$ reductions over 0.2–0.3 kJ/m$^2$ UVC fluences for *E. faecalis* strains ATCC 27285 [63] and DSM 20478 (NCTC 775) [64], albeit in those studies no 4D values were apparent. Overall, *E. faecalis* appeared 2-fold more resistant than *E. coli* to UVC (Figure 2, Table 2).

The Treelium HC devices used in this work incorporated acoustic resonance in the flow, exploiting synergistic effects between hydraulic pressure and sound, with such effects previously reported to achieve 2 and 5 $Log_{10}$ inactivations for *Pseudomonas aeruginosa* and *E. coli*, respectively [65]. Regardless of cavitator type, HC appeared at least 4-fold weaker at controlling the bacterial load of the spiked-deionised water than UVC, offering 0.7 and 1.7 $Log_{10}$ reductions for *E. coli* and *E. faecalis*, respectively (Figure 3a, Supplementary Materials Figure S3). These reductions were in line with the manufacturer's specifications confirming the usefulness of the devices for maintaining the quality of potable water domestically. Average $Log_{10}$ reductions of $\geq 3$ have been reported for both species (for a review see [66]), albeit for different devices and experimental set-ups [9].

In UVC/HC hybrid configurations, the T-Sonic OM worked additively and synergistically with UVC when disinfecting *E. coli*- and *E. faecalis*-spiked deionised water, respectively (Figure 4b). Although data on the combined effects of the bactericidal action of UVC and HC are limited, hybrid UVC-ultrasound (US) treatments have unravelled synergistic effects of the individual technologies on *E. coli* inactivation [14]. However, direct comparisons between UVC/HC and UVC/US are meaningless because (a) both the cavitational nature and its combined effects with UVC will exert bacterial damage with distinct operational mechanisms, and (b) even similar mechanisms will have individual effects on different strains of the same species [66]. It is likely that *E. coli* aggregation may have compromised the efficiency of both technologies, preventing the emergence of synergistic effects.

The bactericidal effectiveness of the UVC/HC treatments was independent of HC position in the sequence (Figure 4a). However, whilst a single UVC exposure was enough to sensitise *E. coli* cells before HC, at least a triple HC exposure was required to sensitize the cells prior to UVC (Figure 4a). In addition, the increased ORP, O$_2$ saturation, and % DO values for the 5th exposure to hybrid treatments of *E. faecalis*-spiked deionised water relative to UVC suggested the presence of additional oxidants in the water (Figure 5). It is unlikely that the delivery of turbulent water to the devices was responsible for these changes, as O$_2$ saturation and % DO values remained constant for the first and third exposures when UVC and cavitators where individually employed. Thus, changes in the indicator values reflected the generation of reactive oxygen radicals. The increase in deionised water's ORP was 1.4-fold higher when HC succeeded UVC (Figure 5), suggesting that photolytic effects assisted the HC radical generation. Interestingly, cavitator OM-mediated ORP changes were apparent only in the hybrid UVC/HC treatments of spiked deionised water but were not observed in the treatment of tap water, where cavitator PW-mediated ORP increases

were detected (Figure 6). It is likely that the more alkaline pH of the tap water, as well as its distinct chemical content, challenged the operational capacity of the OM cavitator. In future studies, we will investigate the nature of these radicals.

In borehole water a 0.75 $Log_{10}$ inactivation was achieved by HC at 10 L/min (Figure 3b). In addition, the highest HC-mediated reductions were achieved 5 min following the system's washout (Supplementary Materials Figure S8). In agreement with previous works [66], the longer the washing process, the lower the influent's load and the lower the inactivation efficiency. Consequently, the nature of the bacterial load and its low concentration in borehole water have made the clear definition of the PW's inactivation efficiency difficult.

Chemical analyses of borehole water treated with either of the two technologies on site was only associated with a moderate decrease in its nitrate content (Table 4). It is possible that traces of chlorine [67] at the applied UVC-radiation intensity may have encouraged the partial elimination of $NO_3$. Similarly, cavitation-induced radicals have been shown to facilitate nitrate/nitrite reductions in HC reactors under alkaline pH [68]. However, as the samples' nitrite content was at the quantification limits, and since single samples per flow rate were subjected to analyses, we were unable to reach statistically strong conclusions. Previously, a 14% reduction in the $Ca^{2+}$ hardness of tap water by the 5th passage when using the PW cavitator at 15 L/min was observed (Figure S9). A decrease in the $CaCO_3$ and $Ca^{2+}$ hardness of borehole water was also apparent over the 10–14 L/min flow rates (Table 4). It is possible that at the 10 L/min flow oxygen radicals and/or turbulence generated a lower $Ca^{2+}/HCO_3^-$ ratio which promoted aragonite (a 1.5-fold more soluble polyform of calcite) formation [69].

### 4.2. Chemical Disinfection

Chemical analyses corroborated the linear relationship in the production of DBPs with dose and exposure time for both $ClO_2$ [70] and NaClO [71], with chlorite/chlorate and THMs/HAAs predominating $ClO_2$ and chlorination, respectively (Text S10, Table 5). Formation of chloramines at 0.75 mg/L NaClO, implied ammonia/ammonium contributions to the water's chemical demand (Table 5). Overall, the risk in elevated DBP generation in the treatment of borehole water appeared higher for in situ chlorination than $ClO_2$, implying that potentially, 0.3–0.5 mg/L $ClO_2$ would be more suitable in the primary or secondary disinfection of borehole water than NaClO (Table 5). However, since electrochlorination emerged as a more effective microbial inactivation technology in comparison to $ClO_2$, (Figure 9) and since breakpoints for NaClO were in the 0.6–0.9 mg/L range (Figure S7) a dose of 0.75 ppm NaClO would be more effective in the treatment of RO/blended waters. Opting for the lowest effective NaClO dose, having satisfied the chemical demand, particularly in relation to the currently applied chlorine gas dosing, may prevent the selection of oxidant-tolerant microbiota and/or resistant biofilm-formers, which apart from raising risks to public health can compromise the lifespan of the pipe distribution networks [72,73].

For chemical disinfection, doses decayed with time necessitating the use of a kinetic model accounting for the residual concentration change (Supplementary Materials Figures S4 and S5). Fitting of the modified Hom's model to the *E. coli* and *E. faecalis* inactivation data revealed rate constants ranging from 4.47–16.8 $L^n mg^{-n} min^{-m}$ (Table 3), in harmony with previously published values [74]. Similarly, determined minimum exposure times for achieving a 4 $Log_{10}$ inactivation were >4.0 min [75], with longer $C_t$s implying potential resistance emergence [76], as further exemplified by the tailing phase of the kinetic profiles (Figures 7–9). Inactivation of *E. faecalis*-spiked RO water with 0.75 mg/L $ClO_2$ required a longer 4 $Log_{10}$ exposure time than the inactivation of *E. coli*-spiked RO water (Table 3). Overall, the low upward concavity (as determined by the m and *n* scale parameters of Table 3) of the *E. faecalis* $ClO_2$-inactivation curves (Figure 7) relative to those of *E. coli* (Figure 8) suggested that *E. faecalis* was more likely to resist $ClO_2$ treatments over extended exposure to sub-optimal doses.

Interestingly, $ClO_2$ was at least 1.5-fold more effective in the treatment of *E. faecalis* in borehole (pH 8.4) than in RO (pH 7.5) water, with more significant effects apparent for the 0.3 mg/L dose (Table 3, Figure 7). The enhanced performance of $ClO_2$ at more alkaline conditions was also reflected in the complete elimination of the TBC (37 °C) of borehole water by 15–30 min exposure (Figure S10b). The data were consistent with the increased bactericidal effects of $ClO_2$ reported for *E. coli* in alkaline media, relative to acidic and/or neutral conditions [77].

Electrochlorination of *E. faecalis*-spiked RO water revealed inactivation independency on low dose with similar 4 $Log_{10}$ inactivation times, and upward concavity (Figure 7). In contrast, NaClO-based inactivation of *E. coli* in RO water was dose-dependent (Figure 8, Table 3). Regardless of dose, NaClO was more effective at inactivating *E. faecalis* than *E. coli* (Table 3). This is interesting considering that chlorine has been shown to be more effective in Gram-negative bacterial inactivation [78]. However, in the presence of organic matter, chlorine-based *E. coli* inactivation was shown to be more compromised than that of Gram-positive bacteria [78]. It is likely that increased organic matter, and cellular particle and assimilable nutrient availability, following the destruction of the cellular membrane, may have contributed to shielding phenomena [79], preventing *E. coli* cells from experiencing the full effect of the applied treatment.

NaClO decontamination of *E. faecalis*-spiked borehole water was also low-dose independent with similar exposures required to achieve the 4 $Log_{10}$ inactivation (Figure 7). *E. faecalis* NaClO decontamination was more effective in RO water than borehole water, requiring shorter exposures for 4 $Log_{10}$ inactivation (Figure 7, Table 3). This is consistent with the bactericidal activity of NaClO driven by HOCl, the generation of which increases with decreasing pH [18]. Both NaClO doses eliminated the TBC (37 °C) load of borehole water by 15 min, with longer exposures leading to increased bacterial counts (Figure S10a). By 2.5 h available residual for disinfection decreased by 40–50%, making the treatment less effective and giving rise to higher bacterial counts.

$ClO_2$ treatments of *E. coli*-spiked RO water were 10-fold and 5-fold weaker (reduced inactivation rate constants and longer exposures) than gas chlorination and electrochlorination, respectively (Table 3, Figure 9), in agreement with works demonstrating resistance of *E. coli* O157:H7 to $ClO_2$, probably stemming from the indirect oxidative stress effects of $ClO_2$ relative to the more direct stress of HClO/NaClO treatments [33]. Similarly, $ClO_2$ treatments of *E. faecalis*-spiked RO water were 5.5-fold and 9-fold weaker than chlorination and electrochlorination, respectively (Table 3, Figure 9). Overall, NaClO emerged as a more potent bactericidal against *E. faecalis* than both gas chlorination and $ClO_2$.

*4.3. Feasibility Study Outcomes*

Unsurprisingly, when assessing feasibility, the advantages of one technology were the disadvantages of the other. For example, 0.3 mg/L $ClO_2$ could control the natural bacterial load of the borehole water more effectively than NaClO, yet operationally it had the highest cost (Figure 10). In contrast, NaClO was better suited for the treatment of RO water, with high CAPEX (almost identical to UVC), but could potentially generate the same wealth of DBPs as gas chlorination if appropriate provisions are not in place (Supplementary Materials Table S7). Additionaly, NaClO generation was associated with the highest $CO_2$ emissions for electrical consumption per hour (Supplementary Materials Table S8).

UVC, on the other hand, lacked long-lasting residual activity, and although more operational-cost efficient and $CO_2$-emission friendlier than electrochlorination and $ClO_2$ generation, it had a high CAPEX cost. However, since the deterioration of groundwater resources necessitates use of higher RO% in the blends, and the disinfection performance of NaClO was at least as good as gas chlorination, electrochlorination emerged as the top candidate technology for future water disinfection practices. Hybrid treatments involving NaClO and physical or chemical disinfection methods can offer additional DBP control (Table S7) should water chemical demand changes shift the NaClO chemistry towards elevated DBP formation.

### 5. Conclusions

With the aim of improving the quality of Maltese potable water, we have sought alternative-to-gas-chlorination technologies for its treatment and assessed their disinfection efficacy and DBP generation propensity, as well as their implementation feasibility in plant-pilot studies. We concluded that:

- Of the non-chemical disinfection methods tested, UVC exerted a 4-fold stronger bactericidal activity than HC and was 2-fold more effective in the control of the *E. coli* load in deionised water than the control of the more resistant *E. faecalis*.
- Whilst HC failed to achieve a minimum of 2 $\text{Log}_{10}$ inactivation for the tested strains under the set-ups employed in this work, it exerted additive *E. coli*- and synergistic *E. faecalis*-inactivation effects to UVC at 9.5 and 15 L/min flow rates and prolonged exposure, respectively.
- The synergistic *E. faecalis*-inactivation effect to UVC is attributed to HC-mediated oxygen radical generation contributing to oxidative stress that assisted disinfection lethality.
- TDS (9% change), and $Ca^{2+}$ hardness (14% change) reduction, concomitantly followed the radical generation over prolonged contact times, indicating that HC is valuable for hybrid schemes with UVC, for both disinfection enhancement, inorganic/organic UVC-sleeve fouling, and water hardness control.
- Both physical disinfection technologies generated no toxic DBPs likely to compromise the organoleptic attributes of water. However, absence of stable long-lasting disinfection residual for delaying bacterial recovery, and extended exposure to the treatments, are disadvantageous over chemical inactivation.
- The significant CAPEX costs for implementation of UVC in the treatment of Maltese water, in addition to the infrastructural changes required for its accommodation, make adoption of UVC unlikely.
- $ClO_2$ emerged as a better bactericidal than NaClO in the control of the tested bacteria in borehole water (alkaline pH), whereas NaClO disinfection was ideal in the treatment of RO water (closer-to-neutrality pH).
- $ClO_2$-based borehole water disinfection was associated with chlorate production, whereas NaClO-based disinfection shared the same DBP repertoire with standard chlorination. However, the generated DBPs did not exceed the parametric values of the EU directive.
- The overall better disinfection propensity of NaClO (particularly in the control of *E. faecalis*-RO load) ranked the technology as the best alternative-to-chlorine-gas disinfection, despite its significant CAPEX costs, followed by $ClO_2$.

With the possibility of HC-NaClO hybrid schemes further reducing DBP formation, electrochemical NaClO generation offers the advantages of gas chlorination, albeit with less free active residual, and without compromising the drinking water's organolepsis. Based on the obtained data, we can now proceed with more vigorous plant-pilot studies, characterise the sensory properties of the newly disinfected water to those of its gas chlorinated counterpart, and assess its consumer acceptability.

**Supplementary Materials:** The following supporting information can be downloaded at: https://www.mdpi.com/article/10.3390/w15081450/s1, Text S1: Chlorine treatment of Maltese tap water; Text S2: Preparation of bacterial suspensions; Text S3: UVC fluence and minimum exposure time calculations; Text S4: GinaFit kinetic models used in the current study; Text S5: Assessing the goodness of model fitting; Text S6: HC-mediated bacterial inactivation; Text S7: Total standard deviation calculations from individual group variances; Text S8: $ClO_2$ decomposition half-lives; Text S9: Chemical demand; Text S10: Chemical analyses of borehole water samples following the chemical inactivations; Table S1: Administered UVC irradiation doses for the tested flow rates and minimum exposure times ($t_{RMin}$) for laminar flows; Table S2: A comparison of CAPEX and OPEX costs for the implementation of UVC, $ClO_2$, and electrochlorination (NaClO) in groundwater (borehole) treatments; Table S3: A comparison of CAPEX and OPEX costs for the implementation of UVC, $ClO_2$, and electrochlorination (NaClO) in desalinated water (RO) treatments; Table S4: A comparison of CAPEX and OPEX costs

for the implementation of UVC, $ClO_2$, and electrochlorination (NaClO) in reservoir water (blended water) treatments; Table S5: Scoring system for assessing the microbial inactivation efficiency of UVC; Table S6: Scoring system for assessing the microbial inactivation efficiencies of alternative-to-gas-chlorination technologies; Table S7: Provisions for implementation of alternative-to-gas-chlorination disinfection technologies in the current settings; Table S8: Assessment of disinfection technologies in terms of greenhouse gas emissions ($CO_2$ equivalents); Table S9: Determined $ClO_2$ and NaClO half-lives in RO and borehole waters at 19.4 °C; Figure S1: Cartoon of the chlorination set-up used in the disinfection of Maltese potable water; Figure S2: Cartoon of the UVC-405 sita lamp system with its cross-section; Figure S3: Microbial inactivation ($Log_{10}$ ($N_0/N$) by hydrocavitation using the Treelium® T-Sonic OM and T-Sonic PW; Figure S4: First-order $ClO_2$ decay fits on potable water dosed with different concentrations of $ClO_2$; Figure S5: First-order NaClO decay fits on potable water dosed with different NaClO concentrations; Figure S6: Dose and residual $ClO_2$ relationships in potable water; Figure S7: Breakpoint curves of NaClO- and chlorine-treated waters; Figure S8: Dependence of HC inactivation efficiency on total bacterial count (TBC) input; Figure S9: $Ca^{2+}$ hardness measurements on *E. faecalis*-infected tap water following treatments with T-Sonic OM and T-Sonic PW cavitators over different passages; Figure S10: Time course of pH, TDS, concentration of residual and cfu/100 mL TBS (37 °C) in disinfected borehole water effluents.

**Author Contributions:** Conceptualization, V.P.V., D.S., J.M., G.P. and M.P.; data curation, G.P. and J.M.; formal analysis, G.P. and V.P.V.; funding acquisition, V.P.V., D.S., J.M. and M.P.; investigation, G.P. and I.R.; methodology, G.P., D.S., J.M. and V.P.V.; project administration, J.M., D.S., V.P.V. and M.P.; resources, V.P.V., D.S. and G.P.; software, V.P.V. and G.P.; supervision, V.P.V., D.S., J.M., G.P. and M.P.; validation, G.P., D.S. and V.P.V.; visualization, G.P.; writing—original draft, G.P.; writing—review and editing, G.P., D.S. and V.P.V. All authors have read and agreed to the published version of the manuscript.

**Funding:** This research was funded by the Energy and Water Agency (EWA) of the Ministry for Energy, Enterprise, and Sustainable Development of the Maltese Government (EWA 110/20/2/003-C).

**Data Availability Statement:** Not applicable.

**Acknowledgments:** We would like to thank Ede Kossári-Tarnik for completing the preliminary hardness determinations of the analysed water samples.

**Conflicts of Interest:** The authors declare no conflict of interest. The funders approved the quality of the manuscript prior to submission.

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
