# Peer review of "Evaluation of Alternative-to-Gas Chlorination Disinfection Technologies in the Treatment of Maltese Potable Water"

_water, doi:10.3390/w15081450_

Round 1
Reviewer 1 Report
The authors clearly state the objective of the work. This is a topic that can attract the interest of professionals as well.
The methods used, their advantages and possible disadvantages are described very precisely. The developed experimental systems indicate a well-thought-out design.
The presentation of the results is thorough, followable, and convincing with the communication of statistical tests.
In some cases, e.g. for the cost analysis, I would have strongly relied on what is included in the appendices, but unfortunately: "The webpage you are looking for could not be found. The URL may have been incorrectly typed, or the page may have been moved into another part of the mdpi.com site".
Nevertheless, the discussion convincingly proves the correctness of the results and the conclusion.
Author Response
We wish to thank you for your efforts in facilitating the reviewing process. We understand that given the complexity of the manuscript, a certain amount of effort and time were invested, and so we are grateful to you for helping us improve the quality of the submitted manuscript.
Point 1; “The authors clearly state the objective of the work. This is a topic that can attract the interest of professionals as well. The methods used, their advantages and possible disadvantages are described very precisely. The developed experimental systems indicate a well-thought-out design. The presentation of the results is thorough, followable, and convincing with the communication of statistical tests. In some cases, e.g., for the cost analysis, I would have strongly relied on what is included in the appendices, but unfortunately: "The webpage you are looking for could not be found. The URL may have been incorrectly typed, or the page may have been moved into another part of the mdpi.com site. Nevertheless, the discussion convincingly proves the correctness of the results and the conclusion ”.
Response 1; We wish to thank reviewer 1 for their supportive and encouraging comments. We further need to apologise for the absence of the available link to the supplementary section. We were under the impression that the supplementary section would be accessible to the referees regardless of the link (to be released after manuscript acceptance), which further suggests we may have uploaded the file to a wrong section of the submission portal. On re-submission, we will ensure that the supplementary section is uploaded correctly and is available as it should have been.

Reviewer 2 Report
Manuscript ID: water-2319877 - Review
General comments
This paper by Georgios Psakis, David Spiteri, Jeanice Mallia, Martin Polidano, Imren Rahbay, and Vasilis P. Valdramidis is focused on the evaluation of four alternative-to-chlorination disinfection technologies in the treatment of the Maltese potable water. These technologies are the following: 1. ultraviolet C (UVC) irradiation, 2. hydrodynamic cavitation (HC), 3. Chlorine dioxide generation (ClO2) and 4. electrochlorination (NaClO). The authors conducted bacteriological and physicochemical bench-scale studies to assess the bacterial inactivation efficacy, and by-product generation propensity of these 4 technologies. The paper is very interesting and fits very well the scope of this journal. However, there are some points which are missing and thus, the paper lacks some clarity.
Minor Comments
· Abstract & Conclusion
The authors state in the Abstract that they will look for chlorination alternatives, but the conclusion of their study is that chlorination emerged as the most promising technology followed by ClO2 and UVC. Then in the Conclusions they state that electrochlorination will hopefully offer the advantages of standard chlorination. The authors need to add a more robust conclusion in the end of the Abstract section and a more powerful conclusion at the end of the Conclusion section so that the reader is not confused with their findings.
· Introduction:
The Reviewer advises that the authors read the following papers as they are highly relevant to the role of chlorine disinfection in drinking water systems and thus, highly relevant to the discussion of their study.
Fish, K.E., et al., Unchartered waters: the unintended impacts of residual chlorine on water quality and biofilms.
Zhu, Z., et al., Biofilm formation potential and chlorine resistance of typical bacteria isolated from drinking water distribution systems.
· Materials and Methods
The financial analyses that the authors conducted should be moved in the main part of the paper and not be in the Supplementary material. The readers have completely no information about the analysis in the main paper. The Supporting material is only to “support” the main paper, not to completely miss all the information from the main paper.
· Supplementary Material:
The following supporting information (as a single file) can be downloaded at: www.mdpi.com/xxx/s1 This link does not actually work. The authors moved many Text, Figures and Tables in this section. The Reviewer would advise to move some important information to the main paper as they promise to talk about these aspects in the Abstract: a) implementation, b) practicality, c) adaptability, d) integration, e) environment & sustainability, and e) cost & effect. However, these aspects can be found in a major part in the Supporting material. For example, the Tables are 4 in the main paper and 11 (!!) in the Supporting material. This does not make sense to the Reviewer. Similarly, there are 13 important Figures for the story of the study in the Supporting document and only 7 in the main paper. Realistically, most readers do not read the Supporting document because of lack of time. Please consider again the material moved to the Supporting file and make a reasonable decision based on the acceptable limits of the journal. The readers would otherwise find difficult to understand such complicated study without enough explanation and information.
Author Response
We wish to thank you for your efforts in facilitating the reviewing process. We understand that given the complexity of the manuscript, a certain amount of effort and time were invested, and so we are grateful to you for helping us improve the quality of the submitted manuscript.
Point 1; “The authors state in the Abstract that they will look for chlorination alternatives, but the conclusion of their study is that chlorination emerged as the most promising technology followed by ClO2 and UVC. Then in the Conclusions they state that electrochlorination will hopefully offer the advantages of standard chlorination. The authors need to add a more robust conclusion in the end of the Abstract section and a more powerful conclusion at the end of the Conclusion section so that the reader is not confused with their findings”.
Response 1; In the first submitted version of the article, the last abstract sentence stated, “In situ chlorination emerged as the most promising technology….”, not just chlorination. The use of the term “in situ” is indicative of chlorine generated on site, or electrochemically generated chlorine in the form of NaClO. Though both chlorine gas and electrochemically generated NaClO disinfect through hypochlorous acid, we consider them as separate technologies, since gas chlorination is associated with the release of high free available chlorine (FAC) and increased propensity for DBP generation, and NaClO is generated at lower percentages, remains longer active in the distribution systems, and is associated with reduced haloacetonitrile formation. Such clarifications are now included in the main text (lines 78-83).
However, to alleviate any confusion, and to enhance the clarity of the written text, we substituted the term “standard chlorination” with gas-chlorination, and the term “alternative-to-standard chlorination” with “alternative-to-gas chlorination” or “alternative-to-chlorine gas disinfection”. It now becomes clearer that in this work we compared various technologies and/or alternatives to gas chlorination.
Additionally, in the conclusion section, the first sentence of the last paragraph (line 831) was changed to “With the possibility of HC-NaClO hybrid schemes in further reducing DBP formation, electrochemical NaClO generation offers the advantages of gas chlorination, albeit with less free active residual, and without compromising the drinking water’s organolepsis”. We now hope that any obscurities have been eliminated.
Point 2; “The Reviewer advises that the authors read the following papers as they are highly relevant to the role of chlorine disinfection in drinking water systems and thus, highly relevant to the discussion of their study.
Fish, K.E., et al., Unchartered waters: the unintended impacts of residual chlorine on water quality and biofilms.
Zhu, Z., et al., Biofilm formation potential and chlorine resistance of typical bacteria isolated from drinking water distribution systems.”.
Response 2; We thank the reviewer for bringing these two very influential articles to our attention. We have referenced both articles but rather than utilizing the outcomes of the conducted works to enhance our introduction we used the information to extend our discussion (lines 720-724).
Point 3; “The financial analyses that the authors conducted should be moved in the main part of the paper and not be in the Supplementary material. The readers have completely no information about the analysis in the main paper. The Supporting material is only to “support” the main paper, not to completely miss all the information from the main paper”.
Response 3; The text with the assumptions for the financial analyses have been shifted from the supplementary section to the main text (section 2.9.1). Section 2.9.1 follows section 2.9, on feasibility.
Point 4; “The following supporting information (as a single file) can be downloaded at: www.mdpi.com/xxx/s1. This link does not actually work. The authors moved many Text, Figures and Tables in this section. The Reviewer would advise to move some important information to the main paper as they promise to talk about these aspects in the Abstract: a) implementation, b) practicality, c) adaptability, d) integration, e) environment & sustainability, and e) cost & effect. However, these aspects can be found in a major part in the Supporting material. For example, the Tables are 4 in the main paper and 11 (!!) in the Supporting material. This does not make sense to the Reviewer. Similarly, there are 13 important Figures for the story of the study in the Supporting document and only 7 in the main paper. Realistically, most readers do not read the Supporting document because of lack of time. Please consider again the material moved to the Supporting file and make a reasonable decision based on the acceptable limits of the journal. The readers would otherwise find difficult to understand such complicated study without enough explanation and information.”.
Response 4; We wish to apologise to the referees for the absence of the available link to the supplementary section. We were under the impression that the supplementary section would be accessible to the referees regardless of the link (to be released after manuscript acceptance), which further suggests we may have uploaded the file to the wrong section of the submission portal. On resubmission, we will ensure that the supplementary section is uploaded correctly and that the link is accessible.
- Original Table S2, has been moved to the main manuscript, forming Table 1. The reader can now familiarise themselves with the feasibility tool used to address the implementation, practicality, adaptability, integration, environment & sustainability, and cost & effect performance areas used to assess the tested technologies.
- Original Figures S4, and S5 have been moved to the main manuscript, forming Figures 5 and 6, respectively. The readers can now directly observe the main ORP, DO(%), and O2 saturation alterations over the HC treatments of faecalis-spiked deionised water, as well as the alterations in normal potable water, that are mentioned in the main part of the manuscript.
- Original Figure S10, has been moved to the main manuscript to become Figure 10. The readers can now directly review the financial analyses outcomes for implementation of each technology in the treatment of the corresponding water body type.
- Original Table S8 has been moved to the main manuscript, forming Table 6. The readers can now directly review both the point allocation and total scores for the assessed technologies and relate these to the feasibility tool presented in Table 1.
Thus, the total of the available material in the resulting supplementary section has been reduced to 9 Tables and 10 Figures, and the main manuscript material has been increased to include 6 Tables and 10 Figures, resulting in a more balanced distribution.
Implemented changes of the manuscript are marked in red. Additional text imported from the supplementary text to aid cohesion and narrative flow is also marked in red.
